

# Mars-Bench: A Benchmark for Evaluating Foundation Models for Mars Science Tasks

Mirali Purohit[1,3]✉    Bimal Gajera[1*]    Vatsal Malaviya[1*]    Irish Mehta[1*]    Kunal Kasodekar[1]
Jacob Adler[2]    Steven Lu[3]    Umaa Rebbapragada[3]    Hannah Kerner[1]

[1]School of Computing and Augmented Intelligence, Arizona State University, Tempe, AZ, USA
[2]School of Earth and Space Exploration, Arizona State University, Tempe, AZ, USA
[3]Jet Propulsion Laboratory, California Institute of Technology, Pasadena, CA, USA

## Abstract

Foundation models have enabled rapid progress across many specialized domains by leveraging large-scale pre-training on unlabeled data, demonstrating strong generalization to a variety of downstream tasks. While such models have gained significant attention in fields like Earth Observation, their application to Mars science remains limited. A key enabler of progress in other domains has been the availability of standardized benchmarks that support systematic evaluation. In contrast, Mars science lacks such benchmarks and standardized evaluation frameworks, which have limited progress toward developing foundation models for Martian tasks. To address this gap, we introduce Mars-Bench, the first benchmark designed to systematically evaluate models across a broad range of Mars-related tasks using both orbital and surface imagery. Mars-Bench comprises 20 datasets spanning classification, segmentation, and object detection, focused on key geologic features such as craters, cones, boulders, and frost. We provide standardized, ready-to-use datasets and baseline evaluations using models pre-trained on natural images, Earth satellite data, and state-of-the-art vision-language models. Results from all analyses suggest that Mars-specific foundation models may offer advantages over general-domain counterparts, motivating further exploration of domain-adapted pre-training. Mars-Bench aims to establish a standardized foundation for developing and comparing machine learning models for Mars science. Our data, models, and code are available at: https://mars-bench.github.io/.

## 1   Introduction

Over the past few years, foundation models have revolutionized specialized domains such as medical imaging [56, 61], Earth Observation (EO) [41, 72, 2], law [14, 16], and astronomy [45, 59, 78]. These models, pre-trained on large and diverse datasets, offer strong generalization capabilities and enable efficient fine-tuning on downstream tasks with minimal data. The EO community has embraced foundation models in the last 3-4 years, with an explosion of methods, datasets, and benchmarks aimed at improving performance across a wide range of geospatial tasks.

The key driver of progress in these domains has been the development of high-quality, standardized benchmarks. For example, BigBio [24] and MIMIC-IV [37] have accelerated model advancements by providing consistent evaluation protocols for medical applications. Benchmarks like Geo-Bench [43] and PANGAEA [53] have accelerated progress in EO applications by providing a suite of standardized classification and segmentation tasks for evaluating geospatial foundation models. Geo-

---

✉Corresponding Author: mpurohi3@asu.edu
*Equal Contribution

39th Conference on Neural Information Processing Systems (NeurIPS 2025) Track on Datasets and Benchmarks.

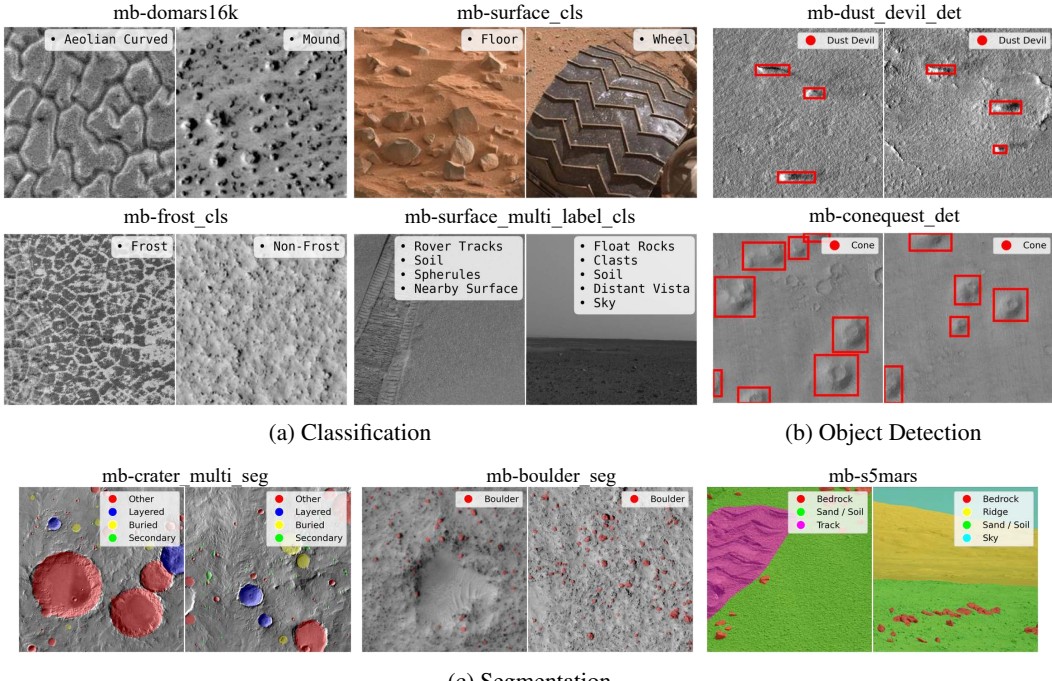

(a) Classification
(b) Object Detection
(c) Segmentation

Figure 1: Representative samples from selected Mars-Bench datasets, from all three task categories.

Bench enables model developers to assess generalization across diverse data sources and use cases, creating a pathway for systematic progress.

However, no such benchmark exists for Martian applications. Machine learning research for Mars science applications thus lags behind other science domains [3]. Although recent studies have presented machine learning solutions for a range of Martian applications, including crater detection [51, 100, 17], landmark classification [94, 88], and cone segmentation [63, 97], these solutions and datasets lack standardization and interoperability. This results in task-specific models or datasets that cannot be easily evaluated as downstream tasks for foundation models or other machine learning advances. This results in limited evaluation of proposed Mars foundation model approaches on 1-2 downstream tasks, limiting the ability to assess model generalization or robustness [86, 93, 91, 26, 65].

This gap is particularly surprising given the richness of available Mars data. Orbiters such as the Mars Reconnaissance Orbiter (MRO) [101] and Mars Odyssey have captured millions of images over the last 20-25 years, while surface rovers like Curiosity and Perseverance have amassed petabytes of high-resolution images. These datasets offer immense potential to study critical questions of planetary science, such as the past presence of water on Mars and the planet's habitability. Yet, the full value of these datasets remains untapped by the ML community due to their lack of standardization, incomplete documentation, and inconsistent formatting for ML workflows.

We introduce Mars-Bench, the first comprehensive benchmark designed to systematically evaluate machine learning models across a diverse set of Mars-related tasks using both orbital and surface imagery. To create this benchmark, we curated and revamped existing datasets, performing quality checks and corrections where necessary and standardizing them in a unified, ML-ready format. The goal of Mars-Bench is to provide a common framework to assess and compare the performance of foundation models on Martian data, facilitating reproducibility and accelerating scientific discovery in planetary science. Our key contributions are as follows:

- **Diverse task coverage:** Mars-Bench includes 20 datasets, summarized in Table 1, spanning three task types: classification, segmentation, and object detection. We also provide a few-shot and partitioned versions of each dataset for evaluation under varying training sample sizes.

- **Scientific relevance:** Mars-Bench covers a wide range of geologic features commonly studied in Mars science, including craters, cones, boulders, landslides, dust devils, atmospheric dust, etc.

These tasks reflect real scientific use cases relevant to planetary scientists and geologists, who co-developed the Mars-Bench. Samples from few Mars-Bench datasets are shown in Figure 1.

- **Comprehensive evaluation:** Since no standardized pre-trained model exists for Mars data, we benchmarked performance using ImageNet-pretrained models under different training settings. We analyzed model behavior with different training set sizes. We also evaluated Mars-Bench using pre-trained EO models as well as proprietary vision-language models, including Gemini and GPT.

- **Code, reproducibility, and baseline models:** We release full code support for all experiments in this paper, along with tools for dataset handling and results visualization. To facilitate community adoption and reproducibility, we also provide well-documented guidelines and publicly release all baseline models evaluated on Mars-Bench. These models can serve as strong starting points for future applications; for example, generating initial global maps of specific geologic features (e.g., cones), which experts can later refine with minimal annotation effort.

## 2    Related Work

Over the past decade, evaluation benchmarks have played a fundamental role in identifying the limitations of existing foundation models, steering their progress in natural language processing (NLP) and computer vision (CV). For instance, general-purpose natural language understanding (NLU) benchmarks [90, 92, 79] have facilitated the development of large language models (LLMs) such as GPT [7], LLaMA [84], and Gemini [83]. Even in specialized domains, including medical [61, 24, 37], legal [23, 27], scientific discovery [50, 11], security [5], and finance [34], various benchmarks have driven progress in building domain-specific foundation models. Thus, development of quality evaluation benchmarks is necessary for building better foundation models.

In the remote sensing domain, Geo-Bench [43] has defined standardized evaluation protocols for a broad set of EO tasks and has quickly become a de facto benchmark. Since its release, Geo-Bench has been used to evaluate most foundation models proposed for EO over the past two years, enabling consistent comparisons across models. Other notable efforts include SustainBench [98], which targets seven sustainable development goals, AiTLAS [18], which aggregates 22 EO datasets focused solely on classification tasks, and PANGAEA [53], which includes 11 evaluation datasets covering diverse satellite sensors.

Despite substantial progress in other domains toward foundation models and dataset benchmarks, no benchmark currently exists for Mars science applications. The absence of a standardized evaluation framework has hindered the development of foundation models (and machine learning solutions more generally) for Mars-related tasks. While specialized datasets exist across different applications, most require significant effort to restructure into an ML-ready format or make interoperable with other datasets. Furthermore, some datasets are not usable without expert guidance from planetary scientists, further slowing progress. To address this gap, we introduce **Mars-Bench**, the first benchmark to facilitate the development and evaluation of foundation models for Mars science tasks.

## 3    Mars-Bench

Mars-Bench was created by curating, organizing, restructuring, and correcting existing Mars science datasets following the design principles explained in Section 3.1. While creating each dataset, our goal was to ensure accessibility and usability and provide task diversity as described in Section 3.2.

### 3.1    Design Principles

**Ease of Use** A key goal was to create an accessible and user-friendly ready-to-use benchmark, supported by standardized data-loading code. We focused on unifying the data format across all tasks to reduce the engineering effort for researchers and practitioners using the dataset. We provide all possible formats in each task if there are multiple common formats. For example, different object detection models may require COCO, Pascal VOC, or YOLO format, so we provide annotations in all three formats to ensure it is easily usable in all cases and reduce time for conversion from one format to another.

**Expert-Validated Corrections** Given the domain-specific nature of Mars science, ensuring high data quality is critical. We conducted expert-driven quality analysis and corrections wherever necessary.

**Classification**

| Name | Observation Source | Geologic Feature | Image Size | # Classes | Train | Val | Test | # Bands | Sensor/ Instrument | Published Year | Cite |
|---|---|---|---|---|---|---|---|---|---|---|---|
| mb-atmospheric_dust_cls_edr | MRO (O) | Atmospheric dust | 100 × 100 | 2 | 9817 | 4969 | 5214 | 1 | HiRISE | 2019 | [19] |
| mb-atmospheric_dust_cls_rdr | MRO (O) | Atmospheric dust | 100 × 100 | 2 | 9817 | 4969 | 5214 | 1 | HiRISE | 2019 | [19] |
| mb-change_cls_ctx | MRO (O) | Surface change | 150 × 150 | 2 | 36 | 10 | 10 | 1 | CTX | 2019 | [40] |
| mb-change_cls_hirise | MRO (O) | Surface change | 100 × 100 | 2 | 3103 | 670 | 670 | 1 | HiRISE | 2019 | [40] |
| mb-domars16k | MRO (O) | Landmark | 200 × 200 | 15 | 11305 | 3231 | 1614 | 1 | CTX | 2020 | [94] |
| mb-frost_cls | MRO (O) | Frost | 299 × 299 | 2 | 30124 | 11415 | 12249 | 1 | HiRISE | 2024 | [20] |
| mb-landmark_cls | MRO (O) | Landmark | 227 × 227 | 8 | 6997 | 2025 | 1793 | 1 | HiRISE | 2021 | [88] |
| mb-surface_cls | Curiosity (R) | Surface | 256 × 256 | 36 | 6580 | 1293 | 1594 | 3 | Mastcam, MAHLI | 2018, 2021 | [88, 89] |
| mb-surface_multi_label_cls | Opportunity, Spirit (R) | Surface | 1024 × 1024 | 25 | 1762 | 443 | 739 | 1 | Pancam | 2020 | [13] |

**Segmentation**

| Name | Observation Source | Geologic Feature | Image Size | # Classes | Train | Val | Test | # Bands | Sensor/ Instrument | Published Year | Cite |
|---|---|---|---|---|---|---|---|---|---|---|---|
| mb-boulder_seg | MRO (O) | Boulder | 500 × 500 | 2 | 39 | 6 | 4 | 1 | HiRISE | 2023 | [62] |
| mb-conequest_seg | MRO (O) | Cone | 512 × 512 | 2 | 2236 | 319 | 643 | 1 | CTX | 2024 | [63] |
| mb-crater_binary_seg | Mars Odyssey (O) | Crater | 512 × 512 | 2 | 3600 | 900 | 900 | 1 | THEMIS | 2012 | [74] |
| mb-crater_multi_seg | Mars Odyssey (O) | Crater | 512 × 512 | 5 | 3600 | 900 | 900 | 1 | THEMIS | 2021 | [44] |
| mb-mars_seg_mer | Opportunity, Spirit (R) | Terrain | 1024 × 1024 | 7 | 744 | 106 | 214 | 1 | Navcam, Pancam | 2022 | [46] |
| mb-mars_seg_msl | Curiosity (R) | Terrain | 500 × 560 | 7 | 2893 | 413 | 828 | 3 | Mastcam | 2022 | [46] |
| mb-mmls | MRO (O) | Landslide | 128 × 128 | 2 | 275 | 31 | 256 | 7 | CTX | 2024 | [60] |
| mb-s5mars | Curiosity (R) | Terrain | 1200 × 1200 | 10 | 4997 | 200 | 800 | 3 | Mastcam | 2022 | [99] |

**Object Detection**

| Name | Observation Source | Geologic Feature | Image Size | # Classes | Train | Val | Test | # Bands | Sensor/ Instrument | Published Year | Cite |
|---|---|---|---|---|---|---|---|---|---|---|---|
| mb-boulder_det | MRO (O) | Boulder | 500 × 500 | 1 | 39 | 6 | 4 | 1 | HiRISE | 2023 | [62] |
| mb-conequest_det | MRO (O) | Cone | 512 × 512 | 1 | 1158 | 167 | 333 | 1 | CTX | 2024 | [63] |
| mb-dust_devil_det | MRO (O) | Dust devil | ∼ 750 × 750 | 1 | 1404 | 201 | 402 | 1 | CTX | 2024 | [28] |

Table 1: Overview of Mars-Bench datasets across all three task categories. To distinguish the benchmarked versions from their original sources, all dataset names are prefixed with "mb-", which indicates Mars-Bench. Observation sources are labeled as O (Orbiter) and R (Rover). Refer to Appendix B.2 for a detailed description and illustrative examples from each dataset.

All segmentation datasets underwent validation by domain experts, and several classification datasets were reviewed and revised through direct correspondence with the original dataset authors. Details on which datasets were corrected or modified are provided in Appendix B.4.

**Dataset Splits** All datasets in Mars-Bench include standardized train, validation, and test splits to facilitate consistent and reproducible evaluation. For datasets that did not originally include predefined splits, we generated them following standard practices. When original splits were available, we preserved them to maintain alignment with prior work. These splits ensure that future methods can be compared fairly and under consistent evaluation settings.

**Cross-Domain Dataset Partitioning** In some cases, we partition datasets based on attributes such as sensor type, data modality, task category, or mission origin. This design choice allows users to analyze model performance across domain shifts, e.g., evaluating cross-sensor or cross-mission generalization by isolating specific factors. Rather than aggregating data into a single dataset, separating them enables experiments in which scientists are often interested, such as how a model trained on one sensor performs on data from another. A more detailed discussion of these partitioning strategies is provided in Appendix B.1.

**Permissive License** All datasets included in Mars-Bench have permissive licenses allowing their re-use in the benchmark. We release the Mars-Bench version of all datasets with a Creative Commons Attribution 4.0 (CC BY 4.0) license, permitting open access and use.

## 3.2 Tasks and Datasets

Mars-Bench offers a diverse collection of 20 datasets spanning three task categories: classification, segmentation, and object detection. Within these categories, the benchmark supports several subtasks, i.e., classification includes binary, multi-class, and multi-label settings, while segmentation includes both binary and multi-class settings. These tasks are constructed from two primary sources of observation: orbiters (satellites) and surface rovers. In total, the benchmark integrates data from 2 Mars orbiters, 3 rovers, and 6 distinct imaging sensors.

The benchmark covers a wide range of scientifically relevant geologic features that are of high interest to the planetary science community and have been extensively studied in prior literature. Mars-Bench was co-developed with expert planetary scientists to ensure its relevance to Mars science. The datasets include geologic features such as boulders, cones, craters, landslides, dust devils, frost, and atmospheric dust. Additionally, multi-class datasets have diverse classes, such as terrain-related classes (e.g., soil, sand, rock, bedrock), landmark-specific features (e.g., Swiss cheese

terrain, spiders, dark dunes), and surface-related elements (e.g., ground, ridges, rover tracks), as well as rover components (e.g., inlet, dust removal tool, scoop). This diversity highlights the breadth of Mars-Bench in terms of task design, sensor modalities, and variety in geologic features. See Appendix B.2 for a detailed description and illustrative examples from each dataset.

Unlike EO datasets in which many classes, such as airports or farmland, can be annotated at scale via crowd-sourcing, Mars science datasets often require annotation by domain experts in planetary science or geology. This process is highly specialized and time-consuming, sometimes taking months to years for high-quality labeling. As a result, as shown in Table 1, several datasets in Mars-Bench are relatively small in size. By including these small-data tasks, Mars-Bench provides a valuable testbed for research on label-limited scenarios.

### 3.3 Using the Dataset

**Availability** All datasets included in Mars-Bench will be publicly released through both Hugging Face Datasets[1] and Zenodo[2]. Each dataset follows a standardized schema and is accompanied by metadata, documentation, and loading scripts to enable easy integration into ML pipelines.

**Target Audience** Mars-Bench offers a diverse set of benchmarks designed to evaluate and compare the performance of foundation models for Mars-related tasks. It serves researchers developing models for planetary applications as well as those interested in the geologic features and data types represented in Mars-Bench. Mars-Bench is also designed to support the broader computer vision and machine learning communities. Researchers studying distribution shift, generalization, or domain adaptation can benefit from its coverage of underrepresented, real-world geospatial scenarios; similar in spirit to WILDS [42]. By offering datasets with unique imaging conditions and semantics, Mars-Bench enables research beyond planetary science.

**Baseline Models** In addition to datasets and code, we release baseline models for each dataset included in Mars-Bench. We will release the models that currently achieve the best performance on their respective datasets. By making these models publicly available, we aim to lower the barrier for applied research. For example, researchers seeking to generate global maps of features such as cones or craters can use our pre-trained models to produce initial predictions, which can then be refined by domain experts with minimal annotation effort.

**Software Tools** To promote reproducibility and facilitate future research, we release an open-source toolkit that encapsulates the complete Mars-Bench experimental pipeline [3]. The repository includes configuration files and executable scripts that reproduce every experiment reported in this study, while permitting users to vary model architectures, hyperparameters, and data partitions with minimal effort. In addition, the toolkit provides utilities for loading datasets and visualizing both objective metrics and qualitative results at the task level as well as in aggregate.

## 4  Experiments

**Model Selection** For each task category, we select well-established and widely adopted model architectures representative of current best practices. For classification tasks, we evaluate ResNet101 [29], SqueezeNet1.1 [32], InceptionV3 [81], Swin Transformer (SwinV2-B) [49], and Vision Transformer (ViT-L/16) [21] architectures. For segmentation, we use U-Net [75], DeepLabV3+ [8], SegFormer [96], and Dense Prediction Transformer (DPT) [70]architectures. For object detection, we evaluate YOLO11 [71], SSD [48], RetinaNet [47], and Faster R-CNN [73].

**Training Settings** We analyze model performance under three different training strategies: (1) training from scratch with randomly initialized weights, (2) using a pre-trained model as a frozen feature extractor, and (3) full fine-tuning of pre-trained models with all weights trainable. As noted in Section 1, no existing foundation model has been trained specifically for Mars tasks. Therefore, we use models pre-trained on large-scale datasets such as ImageNet (for classification and segmentation) or COCO (for detection) as initialization for transfer learning or feature extraction.

---

[1]huggingface.co/collections/Mirali33/mars-bench
[2]zenodo.org/communities/mars-bench/records
[3]github.com/kerner-lab/Mars-Bench

**Hyperparameter Tuning** Since the performance of deep learning models is often sensitive to hyperparameter choices, we conducted a grid search over several hyperparameter configurations for each model, task, and training type combination. The best-performing setting was selected based on early stopping criteria applied to validation metrics. All hyperparameter ranges and selected values for each configuration are detailed in Appendix C.2 to ensure reproducibility.

## 4.1 Reporting Results

We adopt an identical methodology to [1] and [43] to present our results derived from thousands of experiments. Our objective is to report both task-specific outcomes and aggregated results across all tasks with reliable confidence intervals as recommended by [1]. Specifically, for each combination of model, dataset, and training strategy, we first conduct hyperparameter tuning to identify the optimal settings. Subsequently, we retrain each combination using the selected hyperparameters on seven distinct random seeds, since prior work indicates that results based on only 3–5 random seeds may not be sufficiently robust [1]. We follow the exact evaluation and reporting methodology as in [1] and [43], including IQM computation, bootstrapped confidence intervals, and normalization; detailed reporting setup and metrics are provided in Appendix C.4.

## 5 Results and Analysis

In this section, we present baseline results for all three tasks. We structure our analysis around key research questions, which are addressed in the subsections below.

### 5.1 Which model architecture performs best on Mars science tasks, when pre-trained on natural images?

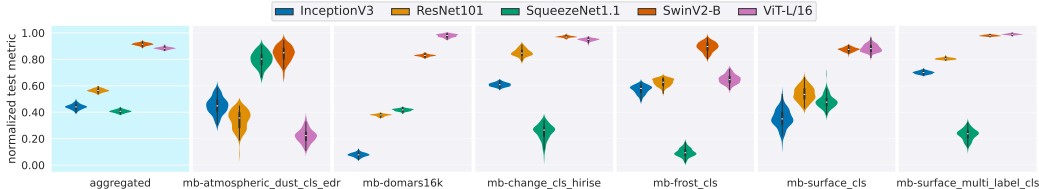

Figure 2: **Classification Benchmark under Feature Extraction setting:** Normalized F1-score of all baselines across six datasets (higher the better). Aggregated plot shows the average over all datasets.

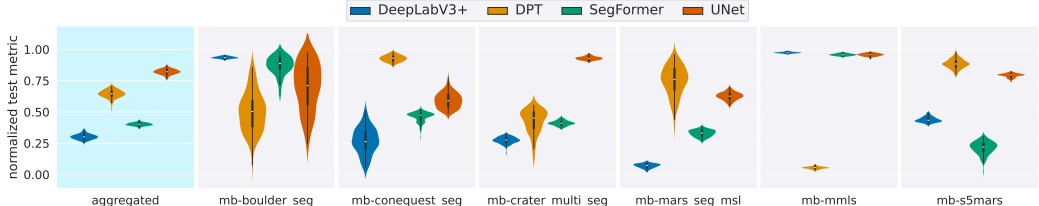

Figure 3: **Segmentation Benchmark under Feature Extraction setting:** Normalized IoU of all baselines across six datasets (higher the better). Aggregated plot shows the average over all datasets.

Figures 2, 3, and 4 show the bootstrapped IQM of normalized performance metric (as defined in Section 4.1) across six classification, six segmentation, and all three object detection datasets and one training strategy (feature extraction with frozen backbone), along with aggregated results. We report the F1-score for classification tasks, IoU for segmentation tasks, and mAP for object detection tasks. For classification and segmentation, the datasets are selected in a way that ensures a diverse set of geologic features. For example, if two datasets cover the same feature type (e.g., landmarks), we report results for only one of them. Additional results, including those for alternative training regimes and other datasets, are reported in Appendix D for all datasets spanning the three tasks..

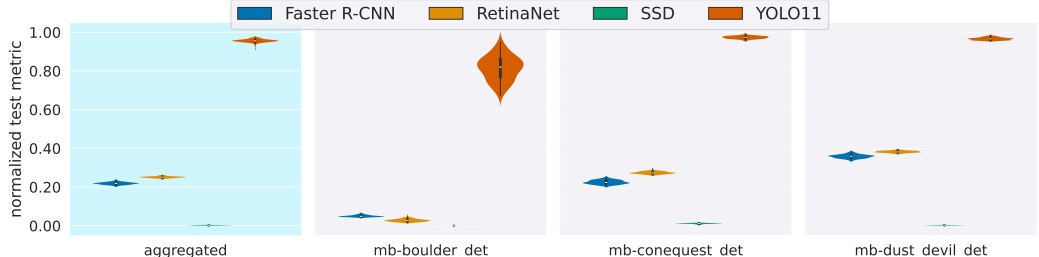

Figure 4: **Object Detection Benchmark under Feature Extraction setting:** Normalized mAP of all baselines across three datasets (higher is better). Aggregated plot shows the average over all datasets.

In classification tasks, SqueezeNet1.1 consistently underperforms relative to other architectures, likely due to its small parameter count. In contrast, ViT-L/16 and SwinV2-B Transformer exhibit competitive performance, with both showing strong generalization across datasets. Notably, some models display narrower confidence intervals than others, suggesting they are more stable and better suited to specific tasks.

For segmentation, U-Net achieves the highest overall performance despite having a relatively wide confidence interval in some datasets. It outperforms both transformer-based models (SegFormer and DPT) on nearly all datasets as well as in aggregate metrics. The DPT model, in particular, shows highly unstable results with large confidence intervals, making it less reliable. These results suggest that, despite its simplicity, U-Net remains a strong baseline for segmentation tasks in Mars science applications.

For object detection, YOLO11 shows the best performance for all three datasets and even in aggregated results. Detection performance is particularly weak on mb-boulder_det and mb-dust_devil_det.

These challenges are primarily due to several factors:

- The overall dataset size is significantly smaller for all three object detection datasets compared to several classification and segmentation datasets.

- The number of objects per image is low, with many images containing only one or even zero target objects.

- The grayscale nature of the imagery limits visual cues, and low object–background contrast (e.g., in dust devil detection) further complicates learning.

## 5.2 What is the effect of training set size on the performance of each model?

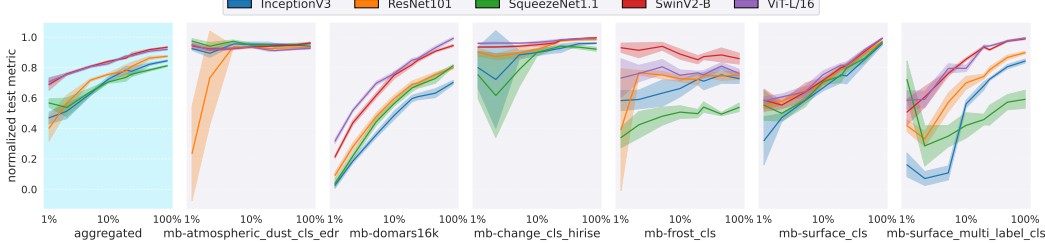

Figure 5: **Classification vs Train size:** Normalized F1-score of baselines with a growing size (from 1% to 100%) of the training set. Shaded regions indicate confidence intervals over multiple runs.

To assess how training set size impacts model performance, we conducted experiments by varying the amount of labeled training data. Specifically, we trained each model using 1%, 2%, 5%, 10%, 20%, 25%, 50%, and 100% of the available training data, while keeping the validation and test sets fixed. For each configuration, we performed multiple runs and report the average normalized test metric, as shown in Figures 5 and 6.

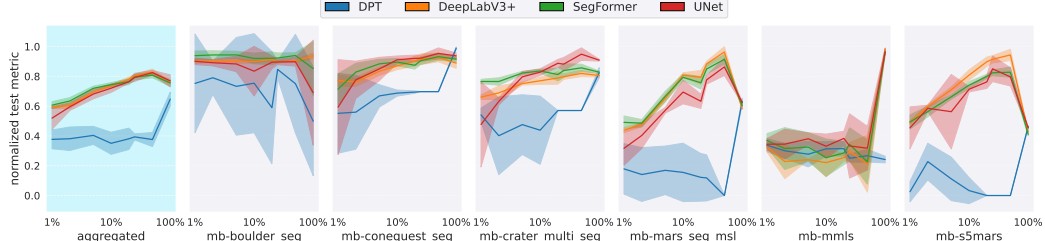

Figure 6: **Segmentation vs Train size:** Normalized IoU of baselines with a growing size (from 1% to 100%) of the training set. Shaded regions indicate confidence intervals over multiple runs.

From the aggregated results, we observe a consistent trend: increasing the training set size generally leads to improved performance in both classification and segmentation tasks. However, dataset-level analysis reveals that the rate of improvement and error margins vary significantly depending on the model and dataset. This shows the differing levels of difficulty among datasets in Mars-Bench, highlighting the benchmark's overall challenge.

In classification, transformer-based models such as SwinV2-B and ViT-L/16 consistently outperform smaller convolutional models like SqueezeNet1.1. In contrast, for segmentation tasks, U-Net outperforms transformer-based models such as DPT and SegFormer across most training sizes. DPT not only shows lower overall performance but also exhibits high variance across runs, as reflected in wide confidence intervals.

### 5.3 How do models that are trained for EO tasks perform on Mars-Bench?

Although there are no published foundation models for Mars orbital or surface imagery, there are many foundation models for Earth orbital imagery. To assess cross-domain generalization, we evaluated foundation models pre-trained on EO data. Specifically, SatMAE [72], CROMA [25], and Prithvi [35] on selected Mars-Bench classification tasks (see Appendix C.1 for experimental details). These models were originally trained on Earth satellite data that vary in geography, scale, and semantics but share the overhead imaging perspective found in many Mars datasets. We compare them to a ViT-L/16 model pre-trained on ImageNet to establish a general-domain baseline (Figure 7).

Although EO pre-trained models performed well on all datasets, the ImageNet pre-trained ViT performed better. One possible explanation is that although ViT is pre-trained on natural images and EO models are pre-trained on satellite data, ViT is pre-trained on 14 million images, while Sat-MAE, CROMA, and Prithvi are pre-trained on 1 million or less than 1 million images. Additionally, diversity in ImageNet, because as discussed in the literature, diversity and/or geo-

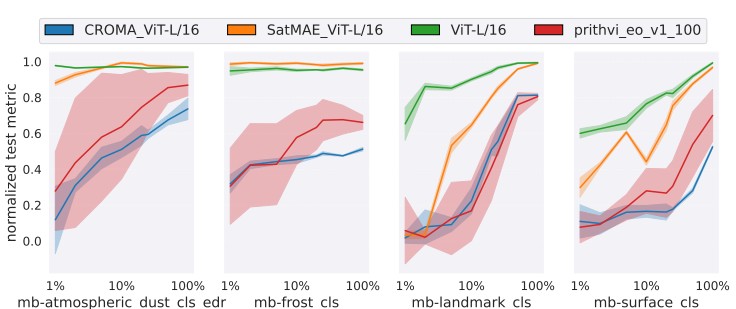

Figure 7: **Classification vs Train size for EO baselines:** Normalized F1-score with a growing size (from 1% to 100%) of the training set. Shaded regions indicate confidence intervals over multiple runs.

graphical coverage of pre-training data can affect the performance of the model [22, 58, 64, 68]. Among EO foundation models, the Prithvi model in particular consistently showed low performance and large error bars. All these results show that, despite EO models pre-trained on satellite data, Earth and Mars orbital imagery differ significantly in ways that likely impact model transferability. For instance, Martian imagery lacks vegetation, water bodies, and human-made structures, which are common in EO datasets. Additionally, Mars exhibits unique geological formations, color distributions, and atmospheric conditions that are totally different than Earth imagery. These domain gaps suggest

that while EO-pretrained models can offer a reasonable starting point, foundation models specifically trained on Mars data are likely to yield more robust and generalizable performance for Martian tasks.

## 5.4 How do proprietary VLMs, such as Gemini and GPT, perform on Mars-Bench?

With the rapid advancement of vision-language models (VLMs), such as Gemini [83] and GPT [7], there is increasing interest in evaluating their effectiveness beyond general-purpose tasks. These models, trained on diverse multimodal datasets, have demonstrated strong performance on various open-domain vision benchmarks with minimal supervision. However, their applicability to Mars science, has not been explored. Evaluating VLMs on Mars-Bench provides valuable insight into their ability to generalize to planetary science tasks without domain-specific fine-tuning.

We focused on evaluating the reasoning capabilities of these models by explicitly prompting them with context-rich instructions, rather than relying solely on direct answer generation. We used the Gemini 2.0 Flash and GPT-4o Mini models, both from their May 2025 checkpoints.

We selected six Mars-Bench datasets spanning classification and segmentation tasks. The selected tasks cover a range of geologic features to evaluate how well the models generalize across different scientific concepts. From each dataset, we randomly sampled 500 test images, ensuring the label distribution in the sampled subset matched that of the original dataset.

| Task | Gemini | | GPT | |
|---|---|---|---|---|
| | Accuracy | F1-score | Accuracy | F1-score |
| mb-domars16k | 0.34 | 0.32 | 0.36 | 0.30 |
| mb-surface_cls | 0.43 | 0.44 | 0.42 | 0.41 |
| mb-frost_cls | 0.50 | 0.55 | 0.43 | 0.54 |
| mb-atmospheric_dust_cls_edr | 0.43 | 0.50 | 0.68 | 0.56 |
| mb-crater_multi_seg | 0.37 | 0.41 | 0.49 | 0.51 |
| mb-mars_seg_msl | 0.86 | 0.84 | 0.79 | 0.70 |

Table 2: Performance of Gemini and GPT on Mars-Bench.

This sample size was chosen to balance evaluation fidelity with the computational cost associated with API-based model usage, particularly for GPT. We reformulated segmentation as a multi-label classification task. For both classification and segmentation, we provided system instructions defining each class and prompted the models to predict the relevant classes for each image. Full prompts and system instructions for all tasks are included in Appendix E.

Both Gemini and GPT achieved reasonable performance on some tasks, but their results are inconsistent across datasets (Table 2). Notably, both models perform well on the `mb-mars_seg_msl` dataset, achieving an F1-score of 0.84 (Gemini) and 0.70 (GPT). This dataset involves terrain segmentation with classes such as sand, rock, and sky, classes that are also common in natural images and likely well-represented in the models' pre-training data. In contrast, performance drops significantly on datasets such as `mb-crater_multi_seg` and `mb-domars16k`, which require identification of fine-grained geologic structures like crater types and Martian landmarks.

With this, we also conducted experiments on smaller vision-language models (CLIP [67], SigLIP [85], and SmolVLM [52]), and these models also show similar trends observed for Gemini and GPT (see Appendix F for details). Our results suggest that current VLMs lack sufficient specialized knowledge for accurate interpretation. As noted in Section 3.2, many of these tasks demand domain expertise. These findings highlight the gap between general-purpose vision-language capabilities and the needs of Mars science, further reinforcing the importance of domain-specific model development.

# 6 Research opportunities

Mars-Bench provides valuable research opportunities, not only for the planetary science and remote sensing communities but also for the broader machine learning and computer vision community. Mars-Bench creates the following key research opportunities:

- Mars-Bench will accelerate the development of foundation models specifically tailored to Mars orbital and surface-related tasks by facilitating a systematic evaluation of model performance. It provides essential infrastructure for benchmarking diverse models within a unified framework, mirroring the influential role benchmarks have historically played in other specialized domains.

- The benchmark comprises several challenging datasets that introduce unique complexities to computer vision tasks. For instance, dust devil detection is particularly challenging due to the subtle

contrast differences between dust devils and the Martian terrain. ConeQuest presents difficulties stemming from significant visual variability among cones collected from various Martian regions, challenging models to generalize across high intra-class variance. In addition, many datasets included in Mars-Bench are small-scale and highly imbalanced, i.e., mb-change_cls_ctx, mb-boulder_seg, and mb-boulder_det.

- Mars-Bench significantly expands research opportunities focused on addressing distribution shifts and out-of-distribution generalization. These challenges are closely aligned with contemporary methodological advancements such as those proposed by [33, 95, 42, 87, 12, 69, 22, 54, 66, 82, 9, 31, 77, 57], which emphasize robust model evaluation across diverse domains to enhance real-world applicability and to advance understanding of model robustness, generalization, and failure modes when exposed to out-of-distribution (OOD) data.

## 7    Conclusion

We introduced the first benchmark for evaluating models on a wide range of Mars science tasks using both orbital and surface imagery. Mars-Bench standardizes diverse datasets into a unified, machine-learning-ready format and provides code for fine-tuning and evaluating across classification, segmentation, and object detection tasks. Datasets in Mars-Bench also include a wide variety of geologic features that have been extensively studied in the literature and remain of high interest to the scientific community. We believe that Mars-Bench will drive the development of Mars-specific foundation models, improve generalization across planetary tasks, and open new research directions in planetary science and beyond.

**Limitations**    A key limitation of Mars-Bench is the absence of georeferencing for most datasets. This arises from the fact that the original sources of these datasets do not provide spatial metadata (e.g., latitude and longitude coordinates), mapping the samples to the Martian surface. As a result, it is currently not possible to assess the spatial distribution or coverage of Mars-Bench across different regions of Mars. Lack of georeferencing is a known challenge in remote sensing benchmarks, as it restricts the ability to conduct spatial analysis or regional generalization studies. There are a few exceptions within the Mars-Bench collection that include geolocation information. For instance, the ConeQuest dataset already provides georeferenced samples, and we retain this metadata in our release. Additionally, both crater segmentation datasets (binary and multi-class) were prepared by us from scratch, and therefore also include geolocation metadata. Both crater datasets are derived from the THEMIS sensor; however, the current version is based on an older THEMIS release from 2010. A newer version of the THEMIS dataset (released in 2017) [30] is now available and can be utilized in the future to generate updated versions of these two crater segmentation datasets.

**Acknowledgment:** Part of this research was carried out at the Jet Propulsion Laboratory, California Institute of Technology, under a contract with the National Aeronautics and Space Administration. We acknowledge Mihir Parmar for his support with the proprietary VLM experiments. We also thank Shrey Malvi and Hitansh Shah for their initial assistance in building and curating the Crater datasets.

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

# Appendix

## A Mars-Bench Resources

- Project Page - mars-bench.github.io/
- HuggingFace - huggingface.co/collections/Mirali33/mars-bench
- Zenodo - zenodo.org/communities/mars-bench/records
- Github - github.com/kerner-lab/Mars-Bench
- LeaderBoard - huggingface.co/spaces/Mirali33/Mars-Bench
- Baseline Models - huggingface.co/collections/Mirali33/mars-bench-models

## B Mars-Bench Details

This appendix provides detailed documentation of the Mars-Bench benchmark. We describe the dataset naming conventions used (B.1), task details of each of the 20 datasets included (B.2), our process for preparing few-shot and partitioned versions (B.3), corrections and improvements made to original datasets with expert input (B.4), and finally, a list of relevant datasets that were excluded from this release and the reasons for their exclusion (B.5).

### B.1 Dataset Naming Convention

For datasets with well-established names in the original literature (which are DoMars16k, ConeQuest, MMLS, Mars-Seg, S5Mars), we retain the original names and prepend the prefix "mb-". For other datasets, we adopt a consistent naming scheme based on geologic feature (e.g., atmospheric_dust), followed by task type (e.g., cls for classification), and optionally by sensor or data source (e.g., edr) as defined in Section 3.1. The suffixes cls, seg, and det indicate classification, segmentation, and object detection tasks, respectively. This naming scheme allows for easy categorization and filtering within the benchmark.

### B.2 Dataset Descriptions

This section provides brief descriptions of the 20 datasets included in Mars-Bench, including their task type, targeted geologic features, class structure, and observation modality.

#### B.2.1 Classification

In all classification datasets, the data is organized in a structured format. Each split, train, validation, and test contains subfolders corresponding to each class specific to that dataset. Additionally, we provide an `annotation.csv` file containing metadata for every sample, including the following fields: `file_id` (a unique identifier for each sample), `split` (indicating the data partition), `feature_name` (a 3-letter acronym representing the class), and `label` (the numerical class ID). To aid interpretability, each dataset also includes a `mapping.json` file that maps both the class IDs and acronyms to their corresponding full feature names.

**mb-change_cls_ctx and mb-change_cls_hirise**   These datasets are designed for binary classification of surface changes using temporal image pairs; specifically, one image taken before and another after some time period, from the *same* Martian location. The task involves identifying whether meaningful surface change has occurred and classifying between "**Change**" and "**No change**". The dataset includes two versions based on different sensors: CTX and HiRISE. Unlike standard single-image classification, this task requires forming a composite input from two grayscale images (Figure 8 and 9). Following the approach outlined by Kerner et. al. [40], we adopt the composite grayscale method: the blue channel encodes the "before" image, the green channel encodes the "after" image, and the red channel is set to zero. `mb-change_cls_ctx` is the smallest dataset included in Mars-Bench, in terms of the number of samples. Since the original datasets do not provide standard splits, we generated consistent train, validation, and test sets for both CTX and HiRISE versions.

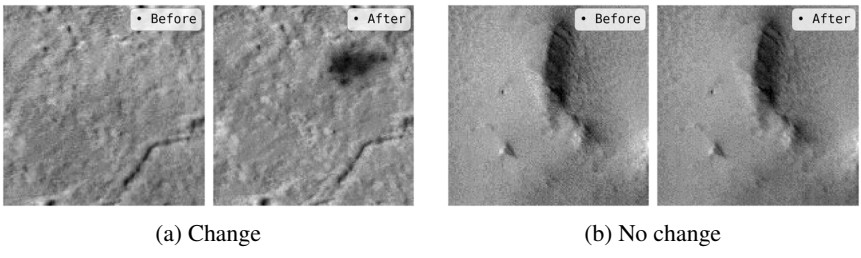

(a) Change        (b) No change

Figure 8: mb-change_cls_ctx

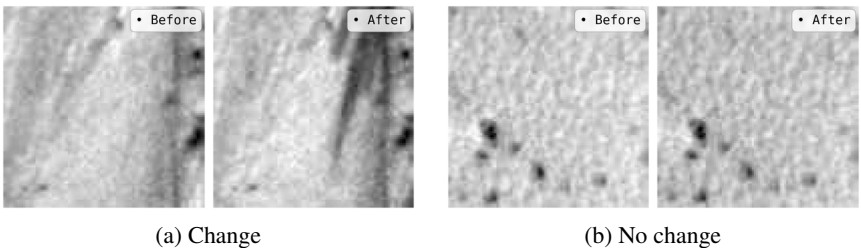

(a) Change        (b) No change

Figure 9: mb-change_cls_hirise

**mb-atmospheric_dust_cls_edr & mb-atmospheric_dust_cls_rdr** These are binary classification tasks focused on classifying between "**Dusty**" and "**Non dusty**" (Figure 10) regions in Mars surface imagery captured by the HiRISE camera on the Mars Reconnaissance Orbiter. The EDR (Experimental Data Record) refers to raw images from the instrument that have not been calibrated or stitched together; while the RDR (Reduced Data Record) is a downsampled or processed version of the EDR, typically used for quick viewing or initial analysis. Both versions are balanced in terms of class distribution and come with predefined train, validation, and test splits.

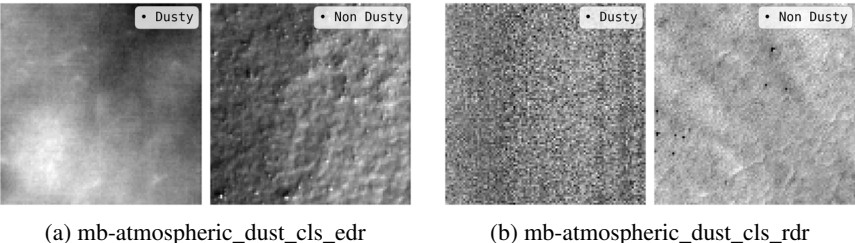

(a) mb-atmospheric_dust_cls_edr    (b) mb-atmospheric_dust_cls_rdr

Figure 10: mb-atmospheric_dust_cls datasets

**mb-domars16k** This is a multi-class classification dataset designed for geomorphologic feature recognition on Mars using imagery from the CTX sensor. It consists of 15 classes (Figure 11) grouped into five thematic categories: (1) **Aeolian Bedforms:** Aeolian Curved, Aeolian Straight; (2) **Topographic Landforms:** Channel, Cliff, Mounds, Ridge; (3) **Slope Features:** Gullies, Mass Wasting, Slope Streaks; (4) **Impact Landforms:** Crater, Crater Field; and (5) **Basic Terrain:** Mixed Terrain, Rough Terrain, Smooth Terrain, Textured Terrain. This is one of the largest and diverse *orbital* datasets in terms of a number of classes. Hence, the dataset presents a unique challenge due to its class granularity, significant variability within classes, and subtle differences between classes, making it valuable for evaluating models on fine-grained classification and generalization. The original version includes train, validation, and test splits; and the dataset is balanced in terms of class distribution.

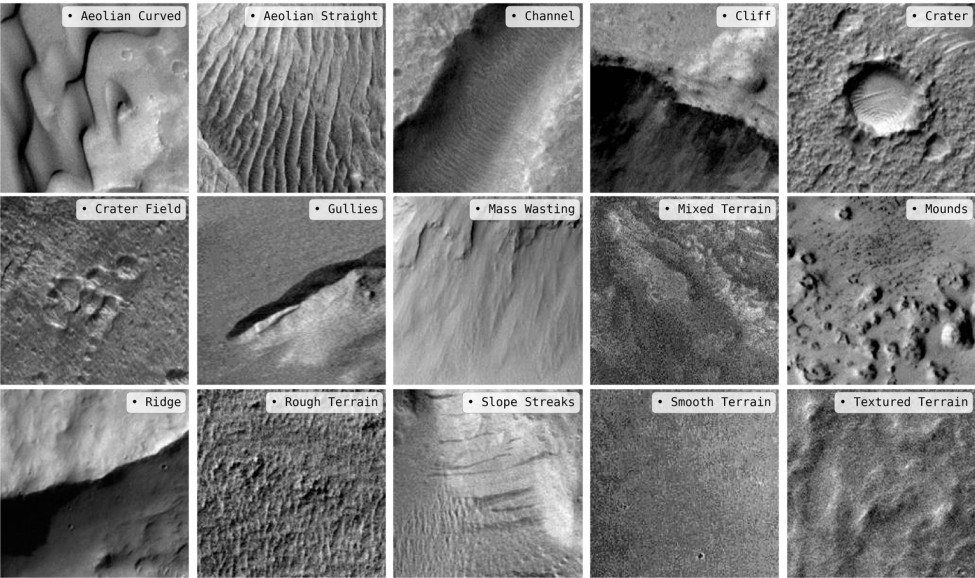

Figure 11: mb-domars16k

**mb-frost_cls**   This is a binary classification dataset designed to detect the presence or absence of surface frost in Mars satellite imagery. The dataset consists of HiRISE image patches labeled as either "**Frost**" or "**Non Frost**" (Figure 12). Among all datasets in Mars-Bench, this is the largest in terms of the number of samples. The dataset is well-balanced in terms of class distribution and includes predefined train, validation, and test splits, as provided by the original authors.

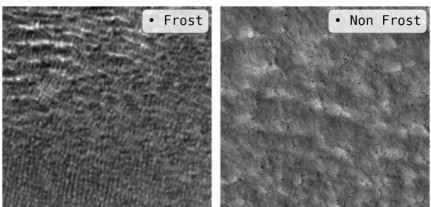

Figure 12: mb-frost_cls

**mb-surface_cls**   This is a multi-class classification dataset consisting of surface imagery captured by the Mastcam and MAHLI instruments aboard the Curiosity rover. It comprises 36 classes (Figure 13), making it the largest and most diverse surface imagery dataset included in Mars-Bench. The dataset was created by combining two previously released versions of the surface classification dataset, following consultation with the original authors (see Section B.4 for details).

The classes span a wide range of surface elements, including rover components, scientific instruments, geologic features, and environmental elements. The full list includes: Alpha Particle X-Ray Spectrometer (APXS), APXS Calibration Target, Arm Cover, Artifact, ChemCam Calibration Target, CheMin Inlet Open, Close-Up Rock, Distant Landscape, Drill, Drill Holes, Dust Removal Tool (DRT), DRT Spot, Float Rock, Ground, Horizon, Inlet, Layered Rock, Light-Toned Veins, MAHLI, MAHLI Calibration Target, Mastcam, Mastcam Calibration Target, Night Sky, Observation Tray, Portion Box, Portion Tube, Portion Tube Opening, REMS-UV, Rover Rear Deck, Sand, Scoop, Sun, Turret, Wheel, Wheel Joint, Wheel Tracks.

Images were labeled using the IDAR tool, incorporating annotations from both domain experts and volunteers. For ambiguous samples, class prioritization rules were applied to assign the most representative label. One part of the dataset ([88]) includes predefined train, validation, and test splits based on Mars sol ranges. For the earlier version [89], which did not include original splits, we created consistent splits before merging with the newer version. As with many real-world planetary

datasets, `mb-surface_cls` is highly imbalanced, with a large proportion of samples belonging to the `Ground` class.

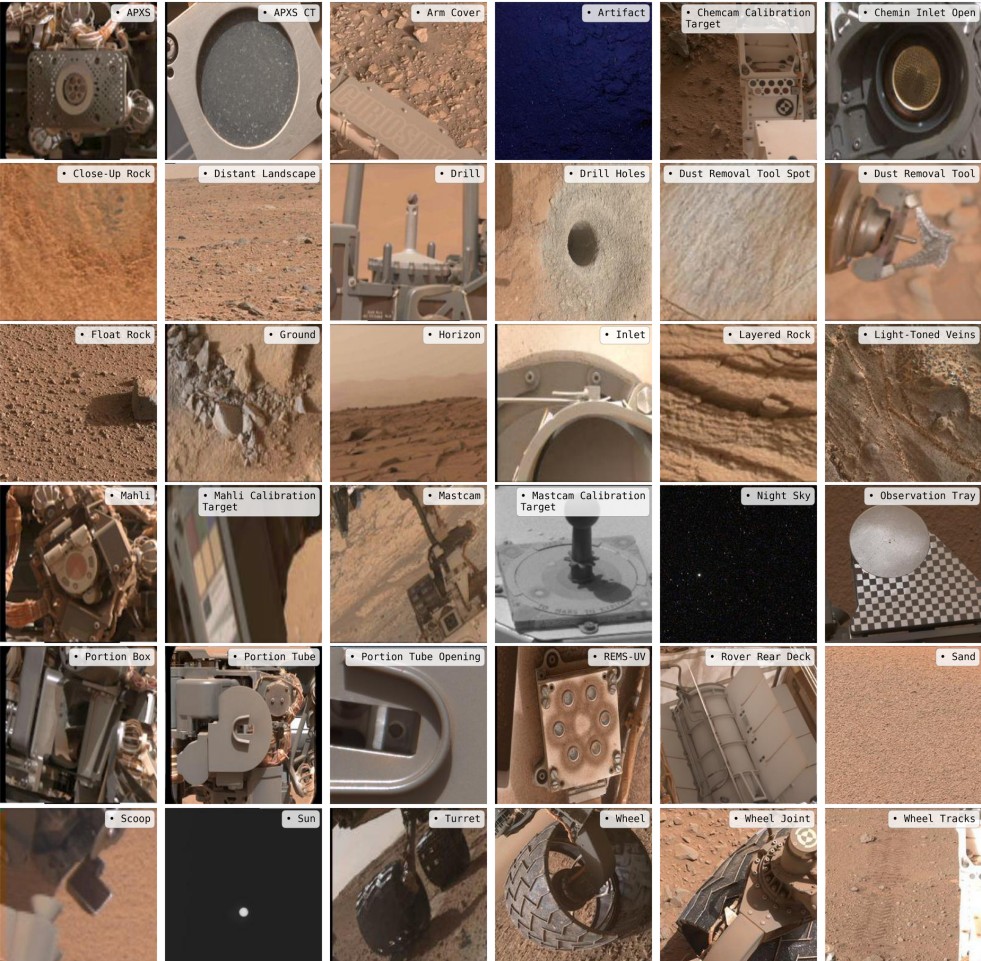

Figure 13: mb-surface_cls

**mb-surface_multi_label_cls**   This is the only multi-label classification dataset in Mars-Bench, based on imagery captured by the Pancam instruments aboard the Opportunity and Spirit rovers. Hence, each image in this dataset can be associated with one or multiple labels (Figure 14). It includes 25 unique classes encompassing a broad range of surface, environmental, and rover-related features. The class list covers geologic and contextual elements such as: RAT Hole, Clasts, Dunes/Ripples, Soil, Rock Outcrops, Close-Up Rock, RAT Brushed Target, Distant Vista, Rover Deck, Bright Soil, Float Rocks, Artifacts, Pancam Calibration Target, Arm Hardware, Round Rock Features, Spherules, Other Hardware, Astronomy, Nearby Surface, Miscellaneous Rocks, Rover Tracks, Sky, Rover Parts, Linear Rock Features, and Soil Trench. The dataset includes pre-defined train, validation, and test splits.

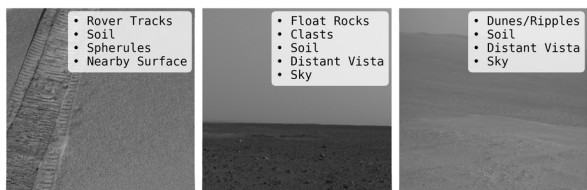

Figure 14: mb-surface_multi_label_cls

**mb-landmark_cls**  This is a multi-class classification dataset derived from orbital HiRISE imagery. It classifies into 8 surface feature classes: **Bright Dune, Crater, Dark Dune, Impact Ejecta, Slope Streak, Spider, Swiss Cheese**, and **Other** (Figure 15). The class distribution is highly imbalanced, with "Other" comprising the majority of samples and Impact Ejecta being the minority class. Landmarks were extracted using a dynamic salience-based method. Labels were generated via a mix of volunteer crowdsourcing and expert validation, with additional calibration techniques applied to improve reliability. The dataset includes predefined train, validation, and test splits.

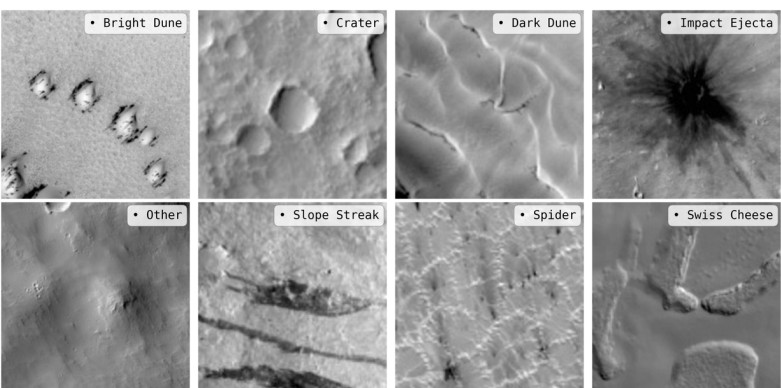

Figure 15: mb-landmark_cls

### B.2.2  Segmentation

All segmentation datasets in Mars-Bench are provided as image–mask pairs, where each mask represents the ground truth labels for semantic segmentation. The masks are encoded as single-channel images, with each pixel assigned a discrete class ID. Across all datasets, the class ID 0 consistently denotes the background. Additionally, we include a `mapping.json` file with each dataset that specifies the mapping between class IDs and their corresponding semantic class names, ensuring clarity and ease of use for downstream tasks.

**mb-boulder_seg**  This is a binary segmentation dataset focused on segmenting boulders on the Martian surface using high-resolution orbital imagery from the HiRISE camera. The dataset comprises manually annotated binary masks indicating the presence or absence of boulders within each image (Figure 16). Boulders were annotated by planetary scientists using precise polygon outlines, ensuring high-quality labels. The dataset originally provides train, validation, and test splits. This is one of the smallest datasets in Mars-Bench with only tens of samples, and that makes it challenging for the computer vision community.

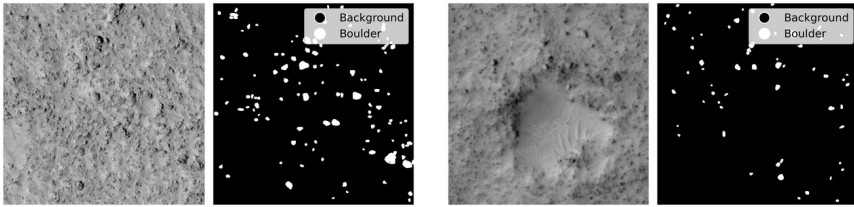

Figure 16: mb-boulder_seg

**mb-conequest_seg**  This is a binary segmentation dataset focused on identifying volcanic cones on the Martian surface using CTX imagery. It was developed to support global mapping and morphologic analysis of small-scale volcanic landforms. The dataset spans three geographically diverse regions on Mars, capturing substantial variation in cone shape, size, and appearance, making it a challenging benchmark for model generalization. Each sample consists of an image and its corresponding binary mask (Figure 17), with all annotations created and validated by expert geologists to ensure scientific accuracy.

Notably, the dataset includes negative samples (images without any cones), which introduces additional complexity by requiring models to correctly predict true negatives rather than detecting cones in every image. We provide metadata indicating which samples are negative, allowing users the flexibility to include or exclude them during training. This information can also be verified using simple image processing techniques. The dataset comes with pre-defined train, validation, and test splits.

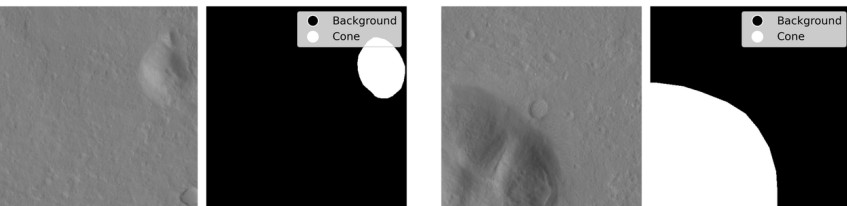

Figure 17: mb-conequest_seg

**mb-mmls** This is a binary segmentation dataset designed to identify landslides on the Martian surface, with a focus on the Valles Marineris region from the CTX sensor. All annotations were manually created by expert geologists, ensuring high-quality, scientifically accurate labels. Each image sample includes multi-modal satellite data comprising 7 channels: RGB (3), Digital Elevation Model (DEM), thermal inertia, slope, and grayscale intensity (in Figure 18, we have visualized grayscale channels). This rich set of modalities captures the complex geomorphology of landslide-prone regions, making the dataset especially valuable for developing and benchmarking robust segmentation models in planetary science. All experiments in this paper utilize only the RGB channels for training and evaluation. The dataset includes predefined train, validation, and test splits to support standardized evaluation.

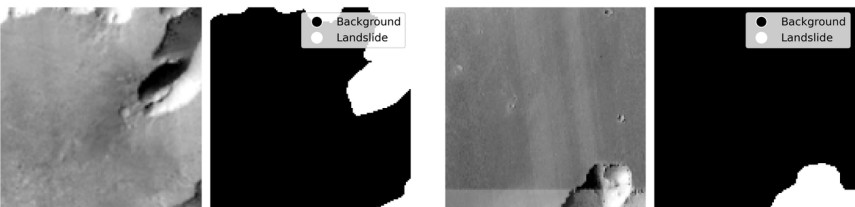

Figure 18: mb-mmls

**mb-crater_binary_seg & mb-crater_multi_seg** These two datasets focus on crater segmentation using THEMIS imagery. `mb-crater_binary_seg` is a binary segmentation dataset that distinguishes crater vs. non-crater regions, while `mb-crater_multi_seg` is a multi-class segmentation dataset with four crater types: Other, Layered, Buried, and Secondary (Figure 19). Craters are crucial for understanding the geological history, surface age, and impact processes of planetary bodies [74]. Moreover, classifying craters into distinct morphological types enables researchers to assess which crater types are more informative for scientific analysis [44].

Although craters are often described as bowl-shaped, they are not always perfectly circular. To address this, we provide annotations in elliptical form, marking the *first* known release of crater segmentation using elliptical geometry, in contrast to the circular annotations commonly used in prior datasets [51, 17, 4]. As the original release consisted only of metadata, we generated the full image-mask dataset using open-source THEMIS data[4]. Due to significant missing pixels near the poles, we restricted the dataset to within approximately $\pm 30°$ latitude of the Martian equator.

To prevent spatial data leakage, we created geographically disjoint train, validation, and test splits. Specifically, images from longitudes $-180°$ to $60°$ are assigned to the training set, $60°$ to $120°$ to the test set, and $120°$ to $180°$ to the validation set.

---

[4]https://www.mars.asu.edu/data/thm_dir_100m/

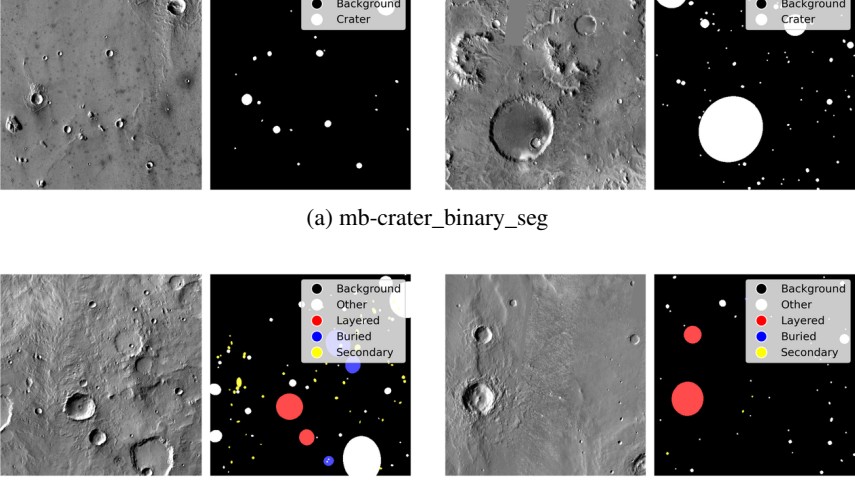

(a) mb-crater_binary_seg

(b) mb-crater_multi_seg

Figure 19: mb-crater_seg datasets

**mb-mars_seg_mer & mb-mars_seg_msl**    These are multi-class segmentation datasets designed
to support terrain understanding on Mars using imagery from two distinct rover missions. The
MSL dataset corresponds to the Mars Science Laboratory (Curiosity) mission and includes imagery
captured by Mastcam, while the MER dataset is sourced from the Mars Exploration Rover missions
(Opportunity and Spirit), using Navcam and Pancam sensors. Both datasets are annotated with six
terrain-related classes: Bedrock, Gravel/Sand/Soil, Rock, Shadow, Sky/Distant Mountains, and Track,
representing typical surface elements observed during rover operations (Figure 20).

While the original datasets were annotated by planetary science experts, we applied additional
refinement in Mars-Bench by consolidating visually similar or ambiguous categories, such as different
granular terrain types, based on expert consultation, aiming to reduce annotation inconsistencies and
improve evaluation reliability (see Section B.4). Since the original datasets do not provide standard
splits, we generated consistent train, validation, and test sets for both MER and MSL versions.

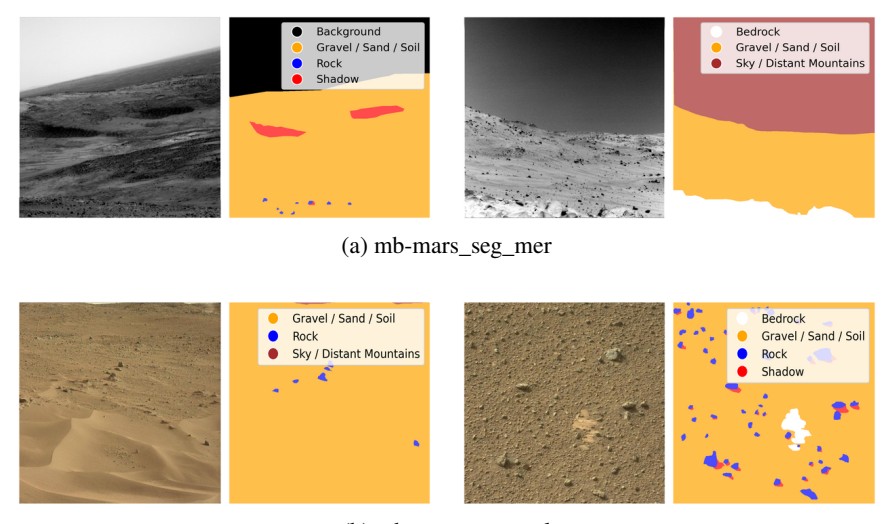

(a) mb-mars_seg_mer

(b) mb-mars_seg_msl

Figure 20: mb-mars_seg datasets

**mb-s5mars**    This is a multi-class segmentation dataset developed to enable semantic understanding
of Martian surface terrain using imagery captured by the Mastcam camera aboard the Curiosity rover.
It contains 8 classes: Bedrock, Hole, Ridge, Rock, Rover, Sand/Soil, Sky, and Track, representing

features commonly encountered during rover-based navigation and scientific exploration (Figure 21). Although the original dataset is reported to be annotated by domain experts, in Mars-Bench we further refine it by merging visually ambiguous classes based on expert analysis to reduce label noise and enhance the robustness of model evaluation (see Section B.4 for details). The dataset includes predefined train, validation, and test splits.

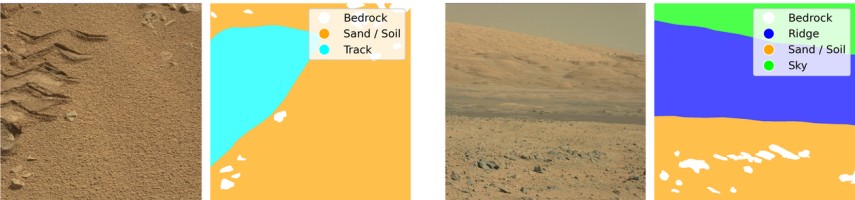

Figure 21: mb-s5mars

### B.2.3 Object Detection

As described in Section 4, all object detection datasets in Mars-Bench are provided in multiple annotation formats to support a broad range of models and frameworks. Specifically, we include annotations in COCO, Pascal VOC, and YOLO formats. This ensures compatibility with most object detection pipelines and reduces the effort and time required for format conversion by end users.

**mb-boulder_det**  This is the object detection version of the Boulder dataset, designed to localize boulders on the Martian surface using high-resolution orbital imagery from the HiRISE camera. Each image is annotated with manually curated bounding boxes that delineate individual boulders (Figure 22), with annotations created by planetary scientists to ensure high-quality and scientifically accurate labels. The dataset includes predefined train, validation, and test splits. This is one of the smallest datasets in Mars-Bench with only tens of samples. Given the small-object nature of the task, this dataset presents a valuable benchmark for evaluating object detection models in low-data regimes, a setting of growing interest in the computer vision community.

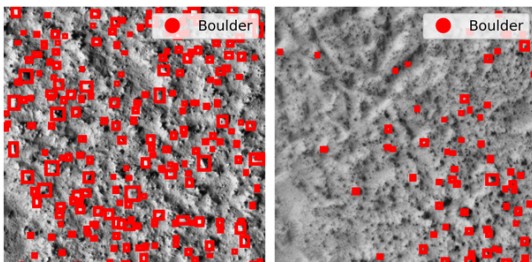

Figure 22: mb-boulder_det

**mb-conequest_det**  This is the object detection version of the ConeQuest dataset, formulated to localize cones on the Martian surface using CTX imagery. It was developed to support global mapping and morphologic analysis of small-scale volcanic landforms. The dataset spans three geographically diverse regions on Mars, capturing substantial variation in cone shape, size, and appearance, making it a challenging benchmark for model generalization. Each sample consists of an image and its bounding boxes (Figure 23), with all annotations created by expert geologists to ensure scientific accuracy.

As the original ConeQuest contains negative samples, we have removed it from this version of the dataset as many detection models do not support training with image samples that do not have any objects. The dataset comes with pre-defined train, validation, and test splits.

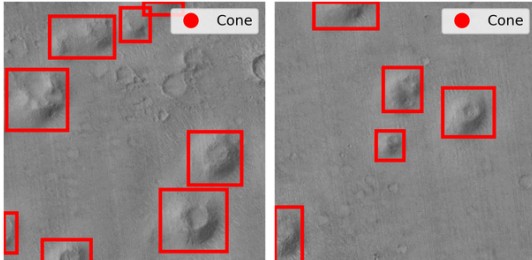

Figure 23: mb-conequest_det

**mb-dust_devil_det**   This task focuses on identifying dust devils in Martian orbital imagery. Dust devils are small-scale, short-lived whirlwinds that play a significant role in Martian atmospheric dynamics, surface modification, and dust transport (Figure 24). The dataset consists of CTX images with manually annotated bounding boxes around visible dust devils. Detecting dust devils presents a considerable challenge due to their faint visibility, small size, and the similarity in texture to surrounding terrain. It includes predefined training, validation, and test splits.

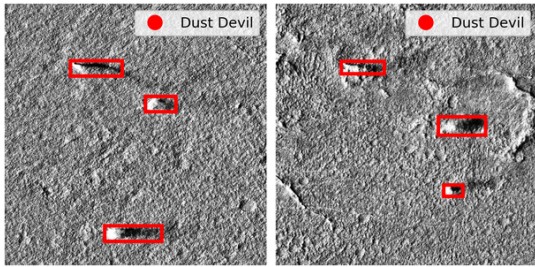

Figure 24: mb-dust_devil_det

## B.3   Few-Shot and Partitioned Data Preparation

To facilitate benchmarking and enable analysis of model performance across varying data regimes, we release both partitioned and few-shot versions of the datasets. These versions are particularly useful for studying how different methods perform with respect to training dataset size and how quickly they reach performance saturation. Importantly, only the training sets are modified in these variants; the validation and test sets remain unchanged across all versions to ensure fair comparisons.

**Partitioned Datasets**   We provide partitioned training sets for all datasets across all task types in Mars-Bench. Specifically, we generate pre-defined subsets using the following proportions of the original training data: 1%, 2%, 5%, 10%, 20%, 25%, and 50%. This results in a total of 131 partitioned datasets. These subsets enable systematic evaluation of how models scale with increasing amounts of data.

**Few-Shot Datasets**   Few-shot versions are provided for all *classification* tasks (except mb-change_cls_ctx) in Mars-Bench. We include the following few-shot configurations: 1-shot, 2-shot, 5-shot, 10-shot, 15-shot, and 20-shot, totaling 54 additional datasets.

For the multi-label classification dataset (mb-surface_multi_label_cls), special care is taken to ensure that each class appears at least the specified number of times in the few-shot setting. For example, in the 2-shot version, every class is represented by at least two instances across the dataset.

**Special Cases and Exceptions**

- For mb-change_cls_ctx, we do not provide few-shot versions, as the original dataset is already very small.
- For mb-boulder_seg, mb-boulder_det, and mb-change_cls_ctx; partitioned datasets are only provided starting from 10%, as smaller subsets resulted in empty or unusable training splits.

These curated few-shot and partitioned datasets are designed to support robust, reproducible research in data-scarce scenarios and facilitate the development and comparison of new methods.

## B.4 Expert-Driven Corrections and Refinements

To ensure the quality, usability, and consistency of the released datasets, we made several corrections and modifications. These changes are aimed at improving clarity, removing redundancies, and aligning the datasets with modern machine learning practices:

- Removal of Augmented Samples: For datasets such as mb-landmark_cls and mb-surface_multi_label_cls, we removed pre-augmented versions of samples that were originally included. We believe it is more appropriate to provide only the raw samples, allowing users to apply their own augmentation strategies using state-of-the-art computer vision techniques. Pre-included augmentations can often be redundant or incompatible with newer workflows.

- Harmonization of Surface and Landmark Classification Datasets: Wagstaff et al. released two versions of the surface and landmark classification datasets [89, 88]. After consulting the authors, we merged the surface classification datasets (released as mb-surface_cls) by:
    - Combining identical classes (e.g., wheel present in both versions).
    - Merging semantically similar classes (e.g., nearby surface and ground).
    - Removing duplicate samples (keeping only one if identical samples existed).
    - Eliminating the ambiguous class Other rover part, which originally served as a catch-all category. The merged dataset now includes a broader and more clearly defined set of surface-related classes.

  For the landmark classification datasets, we retained only the newer version, as recommended by the authors.

- Balancing in ConeQuest Dataset: The original ConeQuest dataset [63] included a significant class imbalance, with only $\sim 12\%$ of samples containing cones (positive samples), and the rest being negative. While the intent was to include broad geographic coverage and true negative learning, we found this imbalance suboptimal for general use.
    - In mb-conequest_seg, we balanced the positive and negative samples across each region while preserving the geographic diversity. We followed the exact methodology described in the original experimental setup.
    - In mb-conequest_det, we excluded all negative samples, as many object detection models do not support samples without any annotated objects.

- Correction of Ambiguous Annotations in Terrain Segmentation Datasets: We performed expert reviews of the terrain segmentation datasets [46, 99] after observing inconsistencies. Experts identified several annotation ambiguities, especially between visually similar classes such as soil, sand, and gravel. These challenges are common in pixel-level annotation tasks due to the fine granularity and visual overlap.
    - In mb-s5mars, we merged the classes sand, soil, and gravel into a single category.
    - In mb-mars_seg_mer and mb-mars_seg_msl, we merged sand and soil.

  These refinements aim to provide cleaner, more consistent datasets that are easier to use, compare, and extend for downstream machine learning tasks in planetary exploration.

## B.5 Excluded Datasets

While our paper includes a carefully curated selection of datasets, there are several others in the literature that we chose not to include for various reasons, detailed below:

- AI4MARS [80]: Upon expert review, we identified annotation errors in this dataset. Since the annotations were produced via crowdsourcing, we found them unsuitable for a highly specialized domain like planetary science, where expert validation is crucial. Consequently, we excluded this dataset.

- Cone Detection (Mills et. al.) [55]: The authors did not release the actual training data. Instead, they shared outputs generated by their own pipeline, global mappings of cones

as bounding boxes with latitude and longitude coordinates. This data is not validated by experts and cannot directly support downstream machine learning tasks.

- Cone Detection (Chen et. al.) [10]: The dataset was not released in a format suitable for machine learning applications. Additionally, no instructions were provided to preprocess or structure the data for ML pipelines, making its use impractical.

- Cone Detection and Segmentation [97]: Although the authors indicated the dataset would be available upon request, we reached out and never received a response, leaving us unable to include the data.

- Novelty Detection [39] and Outlier Detection [38]: These datasets do not fall into the task categories we currently support, i.e., classification, segmentation, or object detection. Furthermore, significant preprocessing would be required. We may consider including them in a future extended version of Mars-Bench.

- Rockfall Detection [6]: Our analysis of the training and test sets revealed a significant number of false negatives (FNs). Although the paper acknowledges this possibility, such inconsistencies hinder reliable model evaluation, especially when FNs are present in the test set, so we excluded this dataset.

- SPOC [76]: The dataset link was not provided in the paper. Upon contacting the authors, we learned that SPOC is an earlier and slightly different version of the AI4MARS dataset [80], and that it is significantly smaller. The authors recommended using AI4MARS instead.

## C  Experiments Details

### C.1  Details of Earth Observation Baselines

For Earth Observation (EO) baselines, we follow the same experimental protocol used for models pre-trained on natural images. Specifically, we perform hyperparameter tuning for each EO model and then train and evaluate the models across all dataset partitions as well as the full training set, using seven random seeds to ensure robust evaluation.

Among the four datasets evaluated with EO-based models, `mb-surface_cls` is an RGB dataset, while the others are grayscale. EO foundation models such as SatMAE, CROMA, and Prithvi are pre-trained on multi-spectral inputs and thus expect input images with multiple channels, often significantly more than standard RGB data. For example, CROMA requires 12 channels for its optical encoder and 2 channels for its radar encoder. As there are multiple versions of these models available, we selected ViT-L from all of them. Since CROMA offers two encoder options, we selected the optical encoder because all the datasets used in our evaluation are from optical sensors. Similarly, for SatMAE, which provides multiple pre-trained encoder choices, we chose the encoder that was pre-trained on the non-temporal subset of the fMoW dataset, as it aligns best with our data characteristics instead of the multi-spectral encoder.

To adapt single-channel or RGB Mars datasets for these multi-spectral models, we replicate the available channels as needed. For grayscale images, the single channel is duplicated to match the required number of input channels. This is a standard practice suggested in prior works and consistent with recommendations in the timm library documentation and repositories such as TorchSeg [15]. While this approach does not introduce new spectral information, it allows for compatibility with the pre-trained architecture without retraining encoders from scratch.

## C.2 Pipeline and Hyperparameters

We provide a user-friendly and scalable training and inference pipeline for classification, segmentation, and object detection tasks. The pipeline supports running experiments via command-line arguments, allowing easy configuration of core parameters such as dataset, model architecture, training type, and hyperparameters.

It includes modular support for logging with options for Weights & Biases (Wandb), TensorBoard, and CSV; model checkpointing; early stopping; and other PyTorch Lightning-compatible callbacks. For reproducibility, we fix the random seed across all relevant libraries and save Hydra configuration files and logs locally. Since Mars-Bench is released on both Hugging Face and Zenodo, the pipeline supports loading data from either platform.

| config | value |
|---|---|
| seed | 0, 1, 10, 42, 123, 1000, 1234 |
| learning rate schedule | w/o, cosine, plateau, step |
| base learning rate | 1e-3, 1e-4, 1e-5 |
| weight decay | 0.05 |
| batch size | 16, 32, 64 |
| optimizer | Adam, AdamW, SGD |
| max training epochs | 50, 100, 200 |
| patience | 5, 10 |

Table 3: Training hyperparameters

As described in Section 4, for each combination of model, dataset, and training strategy, we first perform hyperparameter tuning. We tune the learning rate, learning rate scheduler, weight decay, batch size, optimizer, and maximum number of training epochs. The full search space is listed in Table 3.

We also experiment with different loss functions, summarized in Table 4. For binary classification, we try both a one-node output with binary cross-entropy and a two-node output with standard cross-entropy. For segmentation, we evaluate three loss types: generalized Dice loss, cross-entropy, and a weighted combination of both. We also explore three different weighting schemes. For object detection, we use the default loss returned by each model implementation.

| **Classification** | |
|---|---|
| **config** | **value** |
| criterion | cross entropy, binary cross entropy (only for binary classification) |

| **Segmentation** | |
|---|---|
| **config** | **value** |
| criterion | generalized_dice (square, simple, linear), cross entropy, combined |
| smoothing value | 1e-5 (only for generalized_dice) |

Table 4: Configuration for loss function

## C.3 Number of Experiments

Due to the large number of models, datasets, training strategies, and data splits involved in our benchmark, we summarize here the scale of experiments conducted.

As described in Section 4, we begin by performing hyperparameter tuning for every unique combination of model, dataset, and training type. This includes 13 models, 20 datasets, and 3 training strategies, resulting in 780 hyperparameter tuning runs. Each of these runs was repeated multiple times, depending on the number of configurations in the search space. Once the best hyperparameters were selected, we retrained each configuration using 7 different random seeds to ensure robust and stable performance reporting.

**Classification**  We evaluated 5 models across 9 datasets, under 3 training types and 7 random seeds, totaling **945** classification experiments. In addition, we evaluated all of these combinations on 7 partitioned versions of each dataset, resulting in 6,615 runs. However, for the `mb-change_cls_ctx` dataset, we excluded 1%, 2%, and 5% partitions due to insufficient training samples, which led to the exclusion of 315 experiments. The final count for partitioned classification experiments is **6,300**. We also included few-shot evaluation, using 6 different configurations (1-shot to 20-shot) across all classification datasets except `mb-change_cls_ctx`, which adds **5,040** few-shot experiments.

**Segmentation**   We evaluated 4 models on 8 datasets using 3 training strategies and 7 random seeds, resulting in **672** standard segmentation experiments. We also performed partitioned experiments using 7 training set splits. However, for `mb-boulder_seg`, we excluded the 1%, 2%, and 5% partitions due to extremely limited data, resulting in 252 experiments being skipped. This leads to a total of **4,452** partitioned segmentation runs.

**Object detection**   We ran experiments using 4 models across 3 datasets, again under 3 training strategies and 7 random seeds, totaling **252** standard experiments. We did not perform partition experiments in object detection due to lower performance on the full dataset (more details in Section D.3).

In addition, for EO baselines, we conducted experiments on 4 datasets using 3 EO-pretrained models, evaluated over 7 random seeds and 8 training sizes (7 partitions plus the full dataset), resulting in a total of **672** additional runs.

In summary, we conducted over **19,000 total model runs**, making Mars-Bench one of the most comprehensively evaluated benchmarks for Mars science and planetary vision research. All the experiments were conducted on NVIDIA A100-SXM4 and NVIDIA A30 based on availability on the ASU Sol supercomputer [36].

## C.4   Reporting Results

As mentioned in Section 4.1 and inspired by the methodologies in [1] and [43], we follow a consistent procedure to report results across thousands of experiments. We report outcomes separately for each training setting (random initialization, frozen pre-trained feature extractor, and pre-training with full fine-tuning), which allows for direct performance comparisons across different training settings.

First, we perform hyperparameter tuning for each model–dataset combination, selecting the best configuration based on validation loss using early stopping. Once the optimal hyperparameters are determined, we train and evaluate each model–dataset combination for 7 times with different random seeds, as recommended in prior work [1, 43]. For each combination, we compute the InterQuartile Mean (IQM) by discarding the top and bottom 25% of scores and averaging the remaining values. This approach helps reduce both bias and variance in the reported performance. Before aggregating results across tasks, we normalize the scores within each task to account for differences in scale.

To quantify uncertainty, we perform 1,000 rounds of stratified bootstrapping. In each round, we sample (with replacement) one trial from each dataset, recompute the IQM across all datasets, and build a distribution of IQM values. From this distribution, we calculate 95% confidence intervals. In our final results, we present per-task baselines and overall model performance (aggregated across all tasks) via violin plots. This process is repeated independently for each training setting, and results are reported separately to maintain clarity and consistency.

The results shown in Figures 2, 3, 4, and 5 *in the main paper* are normalized only, without any aggregation. While the main paper and the appendix report both normalized and aggregated results for the feature extraction setting, we also include the corresponding raw results: F1-score for classification, IoU for segmentation, and mAP for object detection. For all other training types, we report only the raw results without normalization or aggregation.

# D   Extended Results

In this section, we present key observations derived from the full set of experiments conducted in this study.

## D.1   Classification Results

Figures 26, 27, and 28 present the classification results (F1-score) for all datasets under feature extraction, transfer learning, and training from scratch settings, respectively. We exclude results for `mb-change_cls_ctx` as it shows negligible variation across different models and training strategies. Overall, feature extraction consistently achieves the highest F1-scores across all models and datasets, followed by transfer learning. Training from scratch performs the worst, with noticeably higher variance. The performance gap between transfer learning and feature extraction is generally smaller than that between transfer learning and scratch, particularly for larger models like SwinV2-B and ViT-L/16.

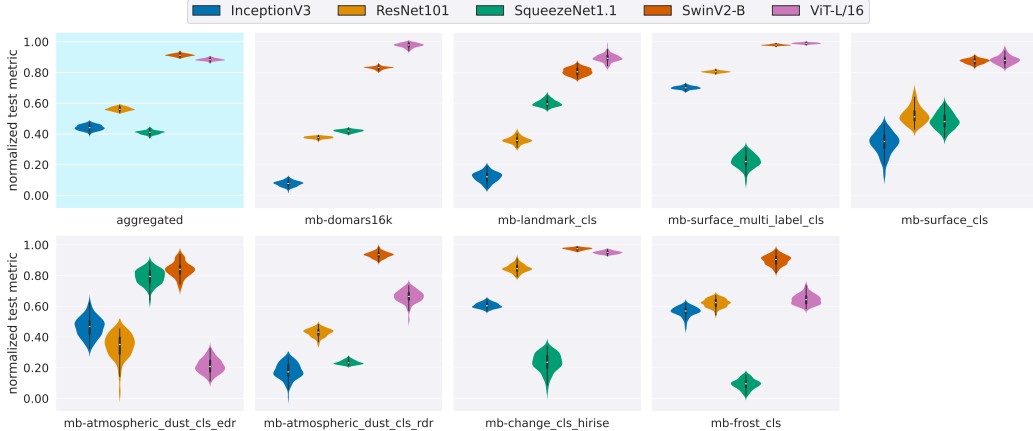

Figure 25: **Classification Benchmark under Feature Extraction setting:** Normalized F1-score of various baselines (higher is better). Violin plots are obtained from bootstrap samples of normalized IQM (Section C.4). The left plot reports the average across all tasks.

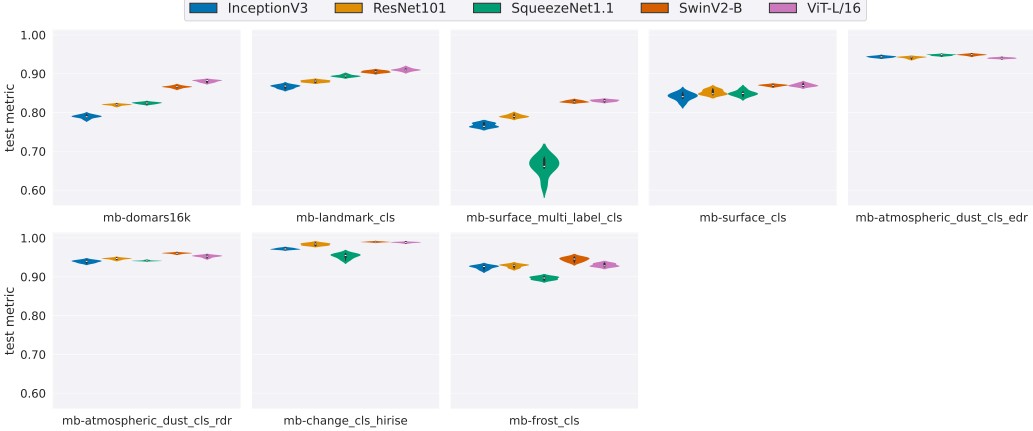

Figure 26: **Classification Benchmark under Feature Extraction setting:** Raw F1-score of various baselines (higher is better). Violin plots represent the distribution of seeds.

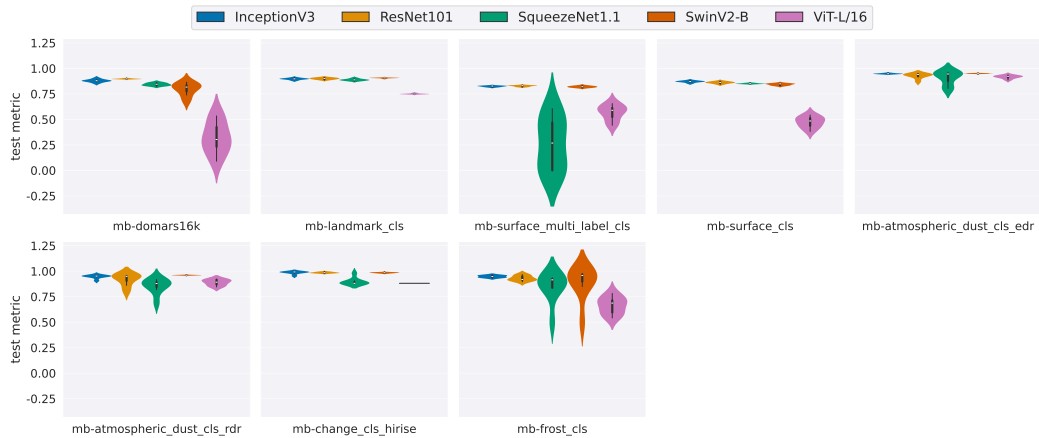

Figure 27: **Classification Benchmark under Transfer Learning setting:** Raw F1-score of various baselines (higher is better). Violin plots represent the distribution of seeds.

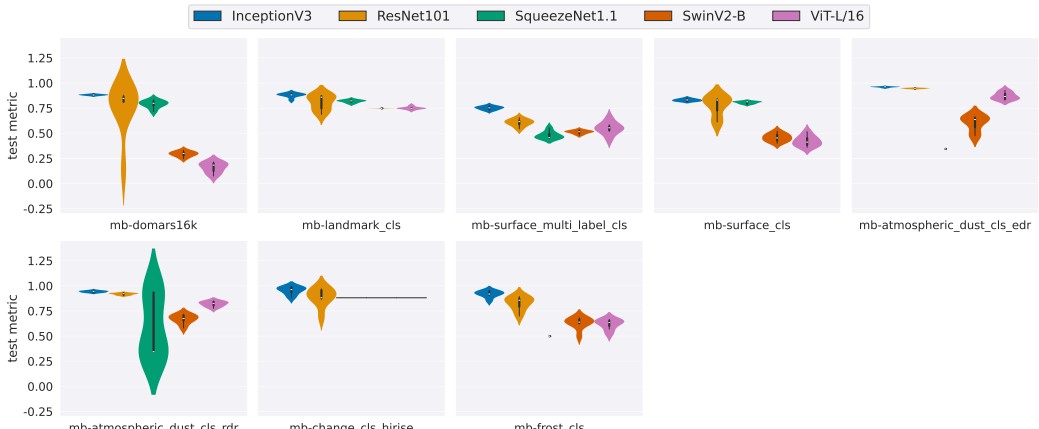

Figure 28: **Classification benchmark with models trained from scratch:** Raw F1-score of various baselines (higher is better). Violin plots represent the distribution of seeds.

### D.1.1 Addressing Data Imbalance

Several datasets in Mars-Bench have significant class imbalance, which can negatively impact model performance, especially for minority classes. In this section, we evaluate the effectiveness of three common strategies; under-sampling, over-sampling, and loss weighting to mitigate data imbalance and assess their impact on classification performance. All experiments are conducted under the feature extraction setting.

- **Under-sampling:** For each dataset, we identify the class with the fewest samples (the minority class) and randomly sample an equal number of instances from all other classes to match this count. This results in a balanced dataset with uniform class distribution, though with a reduced overall training size.

- **Over-sampling:** Rather than directly upsampling all minority classes to match the largest class, which can lead to excessive duplication, we first modestly down-sample the majority class and then apply data augmentation to upsample the minority classes. This approach helps avoid extreme repetition (e.g., scaling a minority class 200×) while still achieving a more balanced class distribution.

- **Loss weighting:** Instead of modifying the dataset directly, we adjust the loss function to give higher weight to minority classes. Class weights are computed inversely proportional

to class frequencies and incorporated into the loss calculation, encouraging the model to pay more attention to underrepresented classes during training.

We applied these techniques across four highly imbalanced datasets from Mars-Bench: (1) mb-landmark_cls, (2) mb-surface_cls, (3) mb-change_cls_ctx, and (4) mb-change_cls_hirise. Experiments were conducted under feature extraction settings using all 5 classification models: SqueezeNet, ResNet, Inception, Vision Transformer (ViT), and Swin Transformer (SwinT). We compare these three balancing strategies against the baseline (standard training without any data manipulation). Results demonstrate how each technique affects overall model performance and performance on minority classes.

|  | mb-landmark_cls | mb-surface_cls | mb-change_cls_ctx | mb-change_cls_hirise |
|---|---|---|---|---|
| **Standard** | 0.84 | 0.77 | 0.78 | 0.86 |
| **Oversampling** | 0.60 | 0.54 | 0.54 | 0.76 |
| **Undersampling** | 0.41 | 0.34 | 0.58 | 0.79 |
| **Loss weighting** | 0.68 | 0.58 | 0.68 | 0.81 |

Table 5: Comparison of class imbalance handling strategies against the standard baseline training, based on weighted F1-score.

- **Overall performance:** From the table 5, it can be observed that while data manipulation techniques help balance classes, they often result in a decrease in overall accuracy compared to the standard setup. Among the techniques, loss weighting generally maintains better performance, but even it does not consistently outperform the standard baseline across all scenarios.

However, the aggregated results do not provide a complete picture of how each technique impacts the performance of the minority and majority classes individually. To interpret these results better, we analyzed the class-wise performance by comparing each data manipulation technique with the baseline (standard setup):

- **Loss weighting:** Same as aggregated results, loss weighting is the most consistent technique in improving minority class performance across all datasets and models, while also maintaining relatively stable performance for the majority class.
  - Example: In the mb-change_cls_hirise dataset using ResNet, the minority class (Change) accuracy improved from 0.17 to 0.48, while the majority class (No change) performance dropped only slightly from 0.96 to 0.94. Similarly, in the mb-surface_cls dataset, the minority class (Arm Cover) improved from 0.00 to 0.50 with ResNet.

- **Undersampling:** Although balancing data, a decreased number of training samples shows a negative effect on performance. Performance for the minority class does not show significant improvement and significantly degrades the performance of the majority class, particularly for transformer-based models, which typically require more data to converge.
  - Example: In mb-landmark_cls, the minority class (Impact Ejecta) improved marginally from 0.00 to 0.03 with Inception, while the majority class (Others) accuracy dropped from 0.93 to 0.35. A similar trend was observed in mb-surface_cls, where the minority class (Arm Cover) improved only from 0.00 to 0.01, while the majority class (Ground) performance dropped from 0.72 to 0.03.

- **Oversampling:** Oversampling shows mixed results. It shows significant effectiveness in transformer-based models but is less consistent in convolutional models.
  - Example: In the mb-change_cls_hirise dataset, ViT showed improvement in the minority class from 0.39 to 0.52, while the majority class remained almost unaffected (0.96 to 0.95). SwinT shows similarly stable behavior across datasets. In contrast, ResNet benefited from oversampling in some datasets (e.g., from 0.00 to 0.62 in mb-surface_cls and 0.17 to 0.35 in mb-change_hirise). However, in SqueezeNet, although it retains the performance of the minority class, it shows performance degradation in the majority class, dropping from 0.84 to 0.51 in mb-surface_cls and from 0.32 to 0.18 in mb-change_cls_hirise.

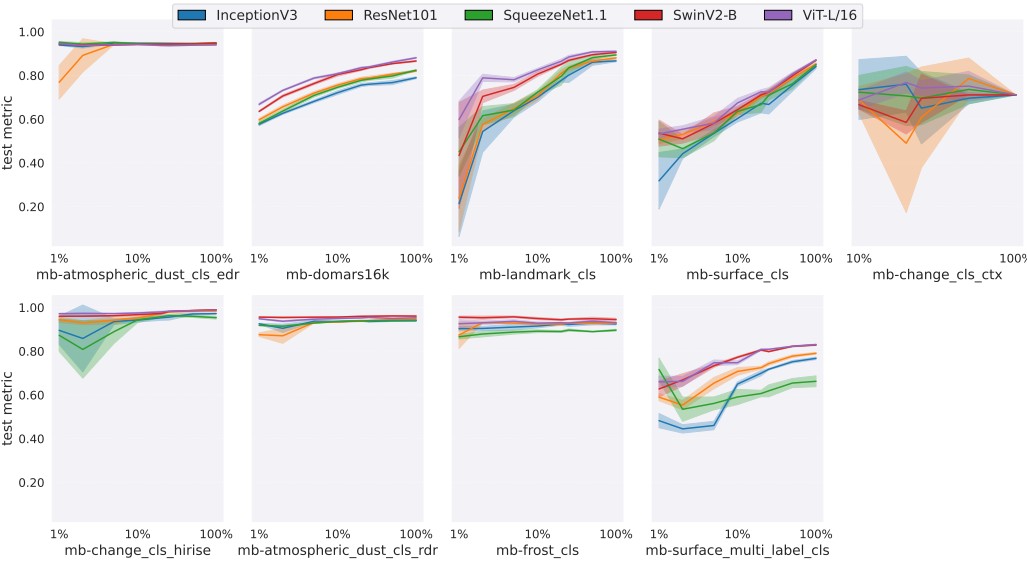

Figure 29: **Classification vs Train size under Feature Extraction setting:** Raw F1-score of baselines with a growing size (from 1% to 100%) of the training set. Shaded regions indicate confidence intervals over multiple runs. Note: Partitions in mb-change_cls_ctx start at 10%.

In summary, all these results indicate that there is no single data manipulation technique that is universally effective across all models and dataset combinations for handling class imbalance. This suggests that the choice of technique depends heavily on the specific characteristics of the dataset and the model being used. We believe this presents a significant opportunity for the community to develop more specialized solutions for imbalanced data in niche domains like planetary science, where class distributions can be highly diverse.

Figures 29, 30, and 31 illustrate how classification performance varies with training set size across the three training strategies. Feature extraction shows rapid performance gains with increasing data and tends to saturate earlier. Transfer learning improves more gradually and typically requires more data to catch up. Training from scratch exhibits slower improvement and higher variability, especially on small datasets like mb-change_cls_ctx, where it often fails to generalize.

Figures 32, 33, and 34 show few-shot learning results across the same training strategies. Consistent with earlier findings, feature extraction significantly outperforms the other approaches across all shot counts, demonstrating strong performance even with as few as 1–2 examples. Transfer learning performs moderately but remains inconsistent across datasets. Training from scratch struggles with very limited data and only starts to improve at 10+ shots, typically lagging far behind the other methods. Note that mb-change_cls_ctx does not have few-shot data due to its already limited dataset size.

## D.2   Segmentation Results

Figures 36, 37, and 38 present segmentation results (IoU scores) across all datasets under feature extraction, transfer learning, and training from scratch settings, respectively. Feature extraction consistently achieves the highest IoU scores across models, with UNet and SegFormer performing particularly well. Transfer learning performs moderately but remains behind feature extraction, while training from scratch shows the lowest performance and highest instability, especially on challenging datasets like mb-conequest_seg and mb-mmls.

Figures 39, 40, and 41 show the impact of training set size on segmentation performance (IoU) for feature extraction, transfer learning, and training from scratch, respectively. As training size increases, feature extraction consistently yields higher and more stable performance across datasets. Transfer learning shows moderate gains but generally lags behind feature extraction. Training from scratch

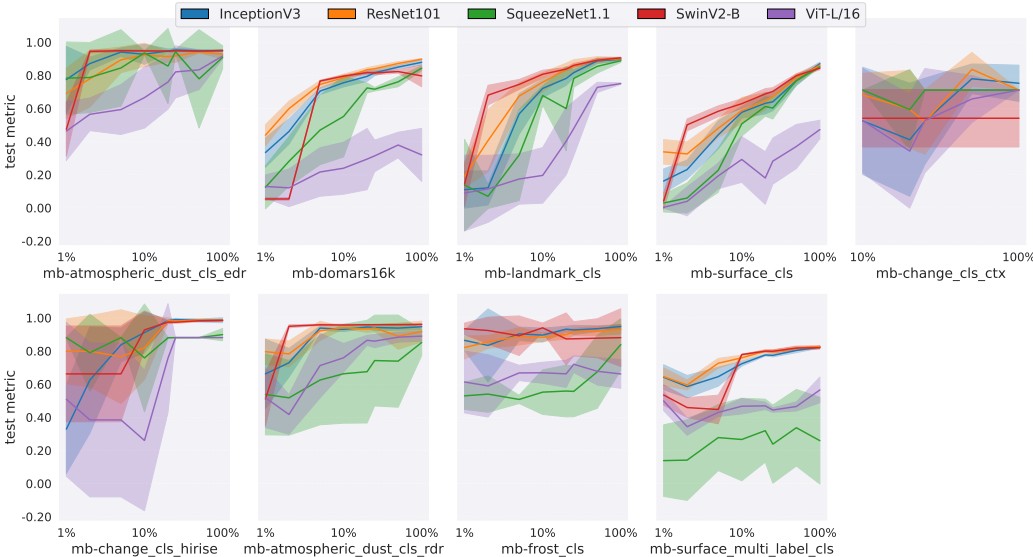

Figure 30: **Classification vs Train size under Transfer Learning setting:** Raw F1-score of baselines with a growing size (from 1% to 100%) of the training set. Shaded regions indicate confidence intervals over multiple runs. Note: Partitions in mb-change_cls_ctx start at 10%.

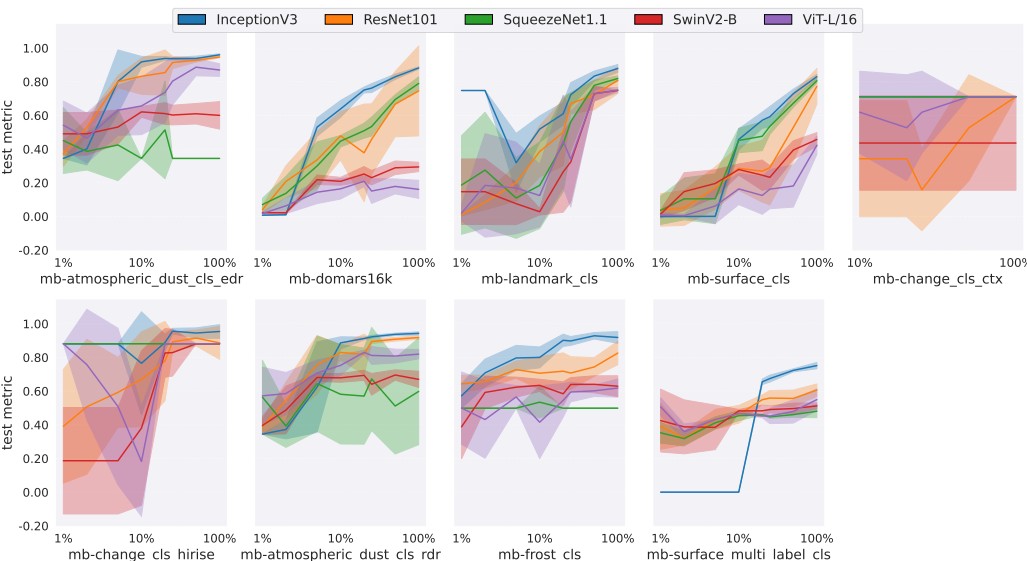

Figure 31: **Classification vs Train size with models trained from scratch:** Raw F1-score of baselines with a growing size (from 1% to 100%) of the training set. Shaded regions indicate confidence intervals over multiple runs. Note: Partitions in mb-change_cls_ctx start at 10%.

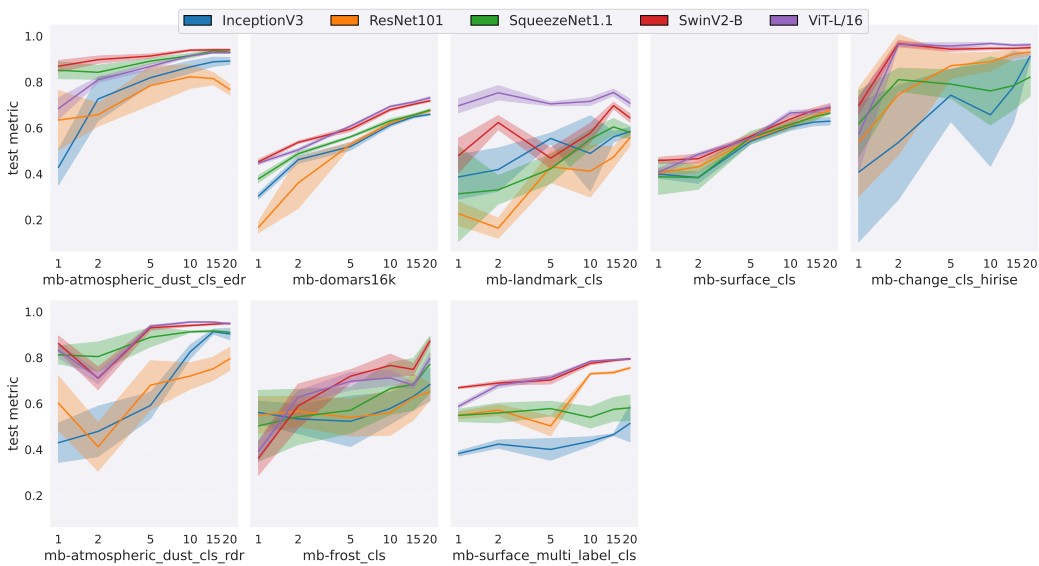

Figure 32: **Classification vs Few-shot under Feature Extraction setting:** Raw F1-score of baselines on few-shot setting. Shaded regions indicate confidence intervals over multiple runs.

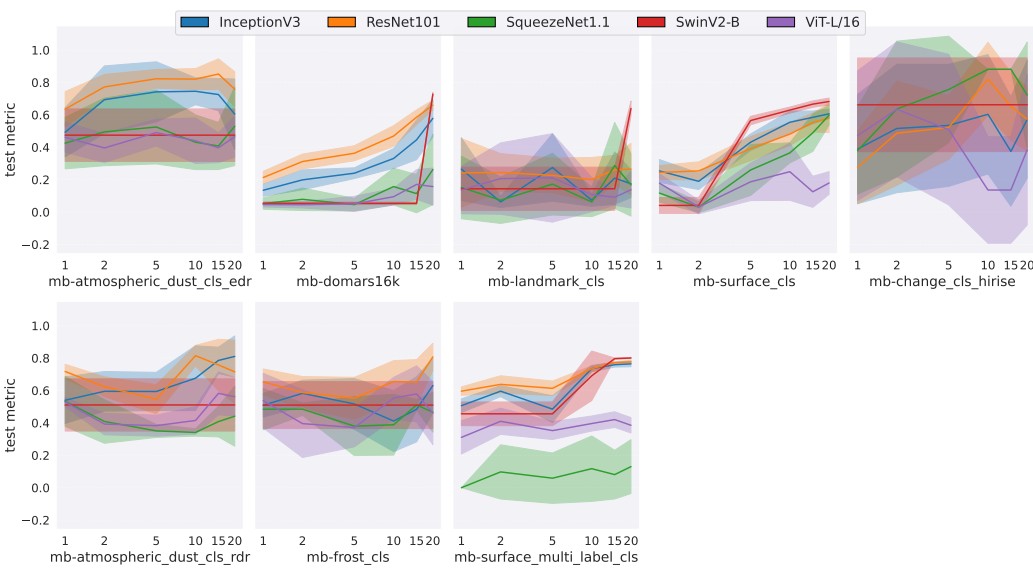

Figure 33: **Classification vs Few-shot under Transfer Learning setting:** Raw F1-score of baselines on few-shot setting. Shaded regions indicate confidence intervals over multiple runs.

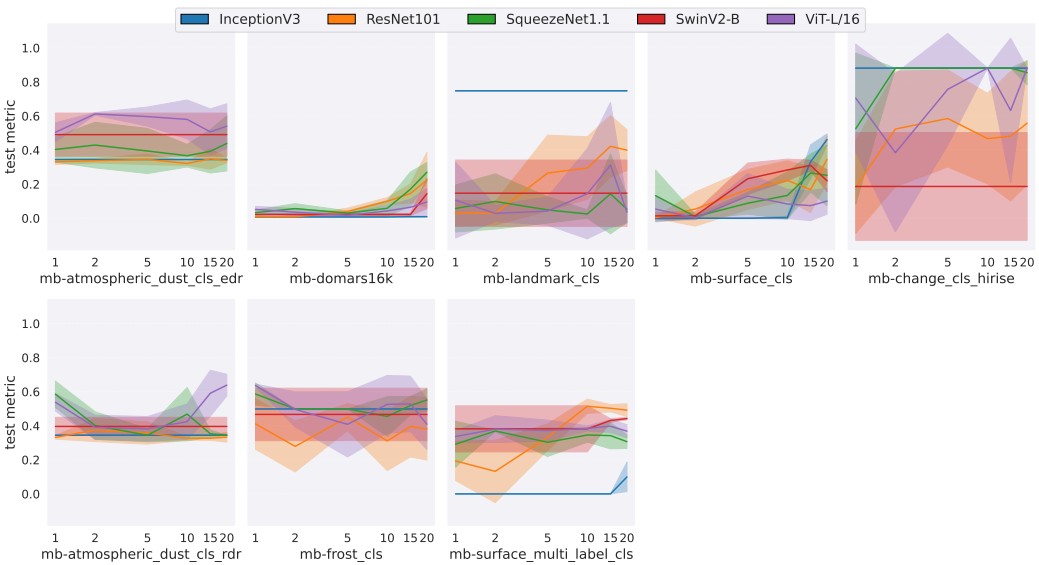

Figure 34: **Classification vs Few-shot with models trained from scratch:** Raw F1-score of baselines on few-shot setting. Shaded regions indicate confidence intervals over multiple runs.

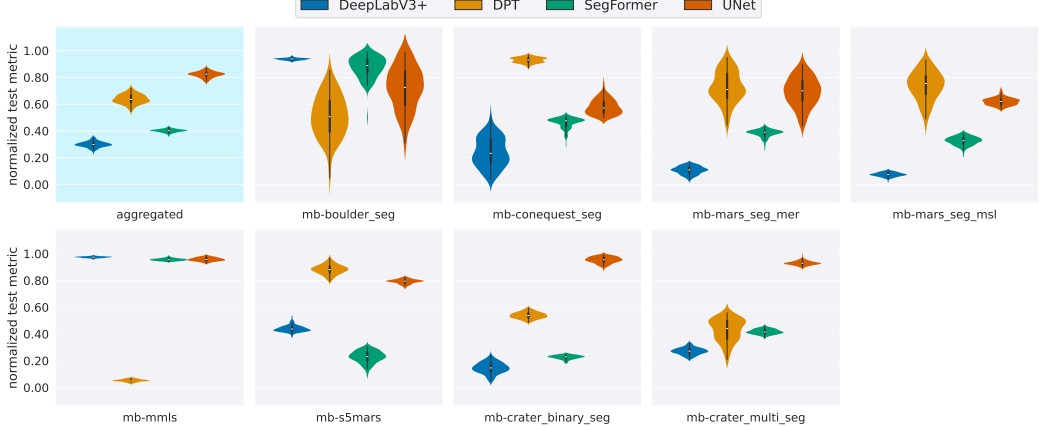

Figure 35: **Segmentation Benchmark under Feature Extraction setting:** Normalized IoU of various baselines (higher is better). Violin plots are obtained from bootstrap samples of normalized IQM (Section C.4). The left plot reports the average across all tasks.

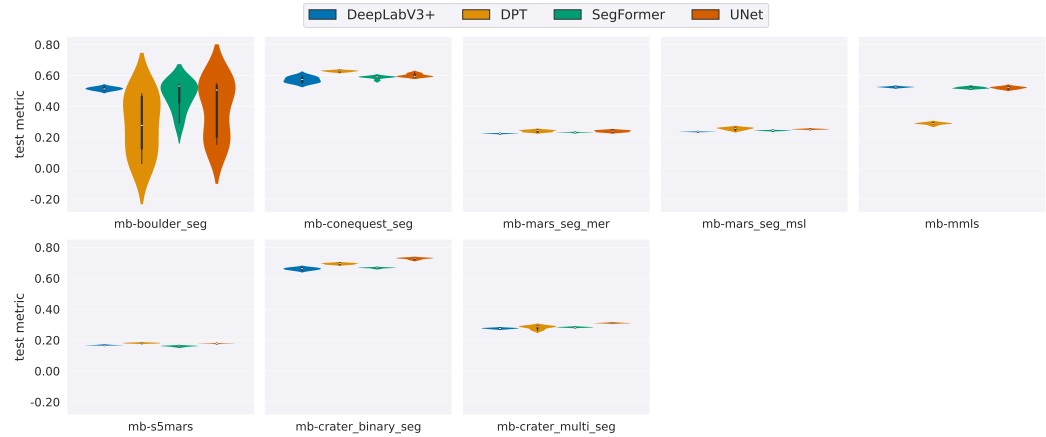

Figure 36: **Segmentation Benchmark under Feature Extraction setting:** Raw IoU of various baselines (higher is better). Violin plots represent the distribution of seeds.

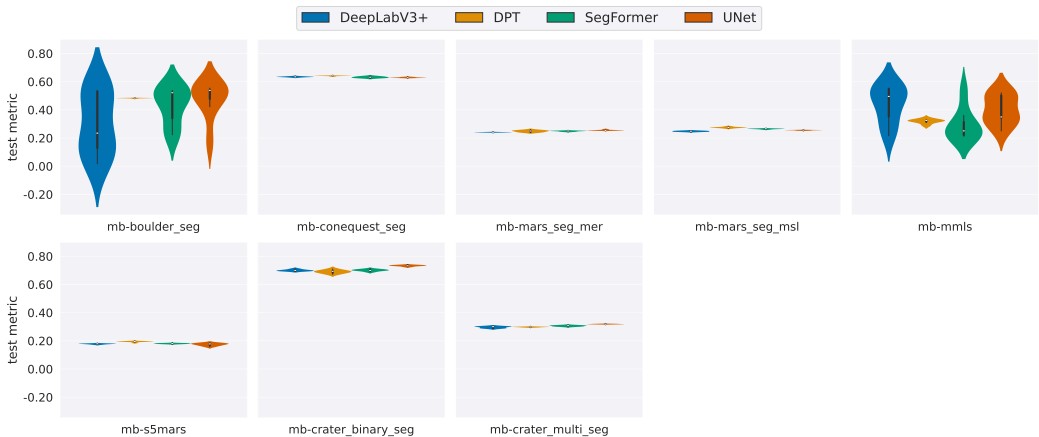

Figure 37: **Segmentation Benchmark under Transfer Learning setting:** Raw IoU of various baselines (higher is better). Violin plots represent the distribution of seeds.

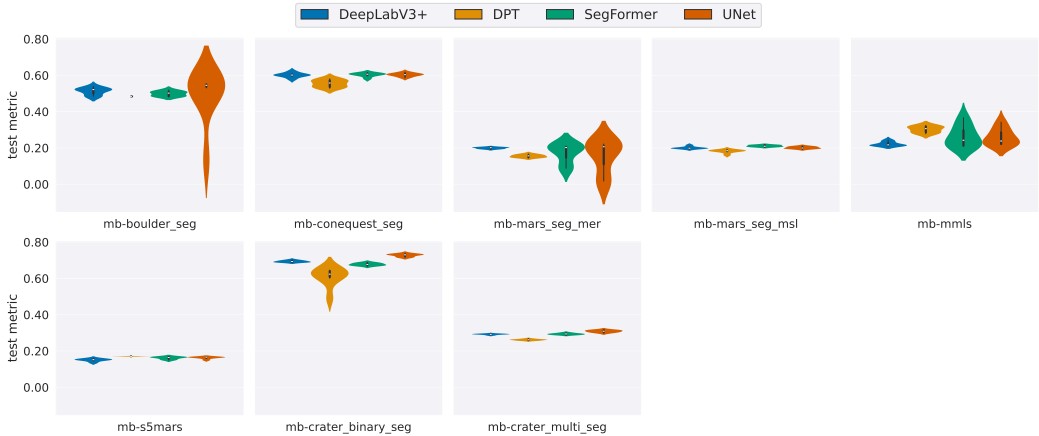

Figure 38: **Segmentation benchmark with models trained from scratch:** Raw IoU of various baselines (higher is better). Violin plots represent the distribution of seeds.

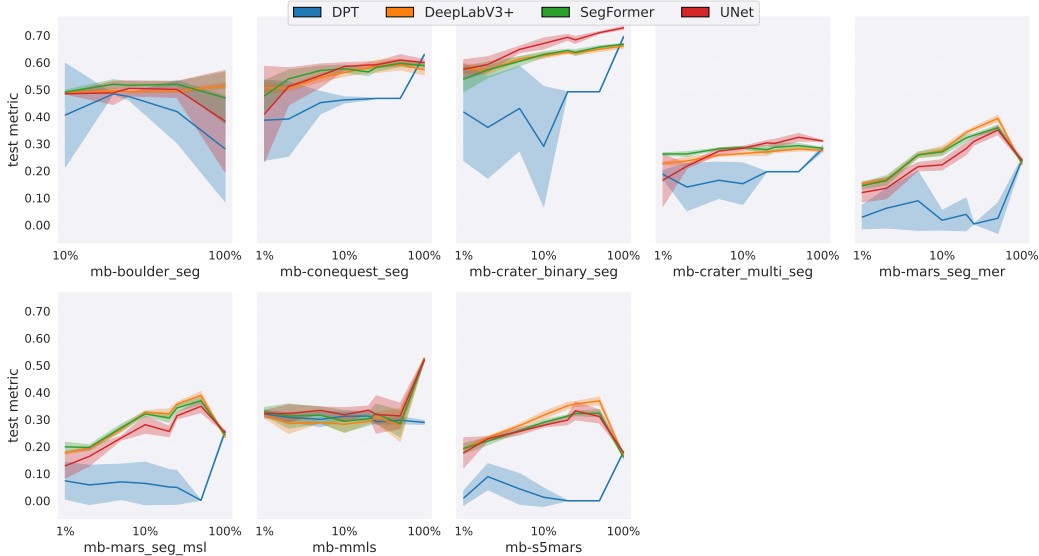

Figure 39: **Segmentation vs Train size under Feature Extraction setting:** Raw IoU of baselines with a growing size (from 1% to 100%) of the training set. Shaded regions indicate confidence intervals over multiple runs.

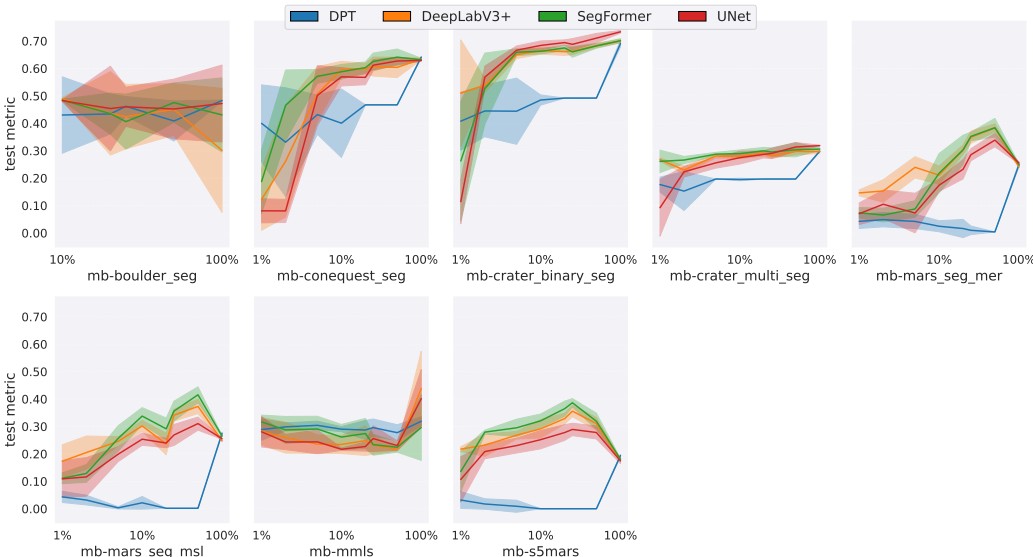

Figure 40: **Segmentation vs Train size under Transfer Learning setting:** Raw IoU of baselines with a growing size (from 1% to 100%) of the training set. Shaded regions indicate confidence intervals over multiple runs.

exhibits the lowest performance and highest variability, particularly on datasets with complex terrain or limited data availability, i.e., mb-mars_seg_mer, mb-mars_seg_msl, and mb-s5mars.

## D.3 Object Detection Results

Figures 42, 43, and 44 present object detection results (mAP) across all datasets under feature extraction, transfer learning, and training from scratch settings, respectively. Feature Extraction achieves the best and most consistent mAP scores across all three detection datasets. Transfer learning performs slightly better than training from scratch, but both show high variability and generally low performance.

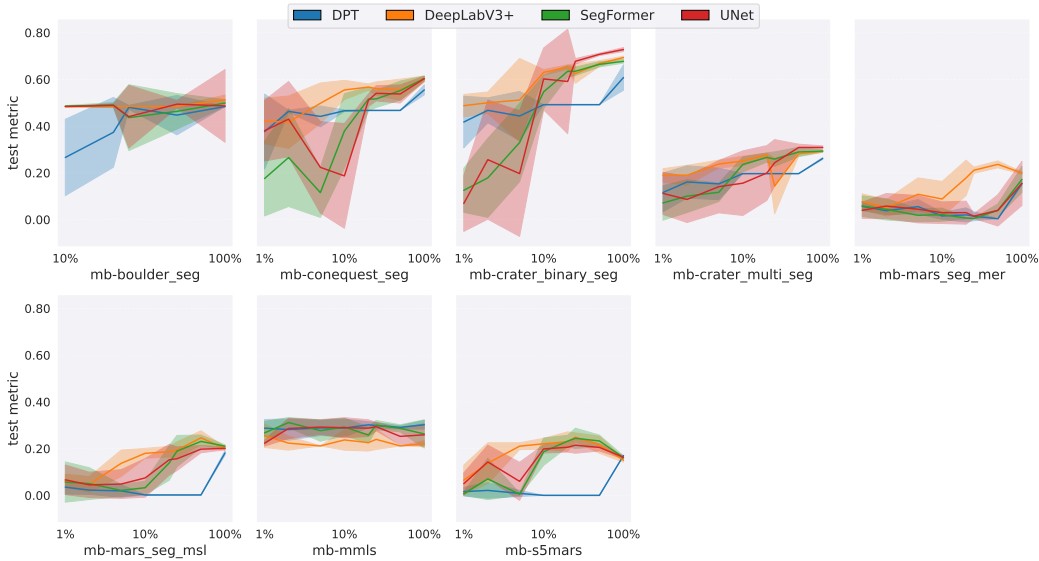

Figure 41: **Segmentation vs Train size with models trained from scratch:** Raw IoU of baselines with a growing size (from 1% to 100%) of the training set. Shaded regions indicate confidence intervals over multiple runs.

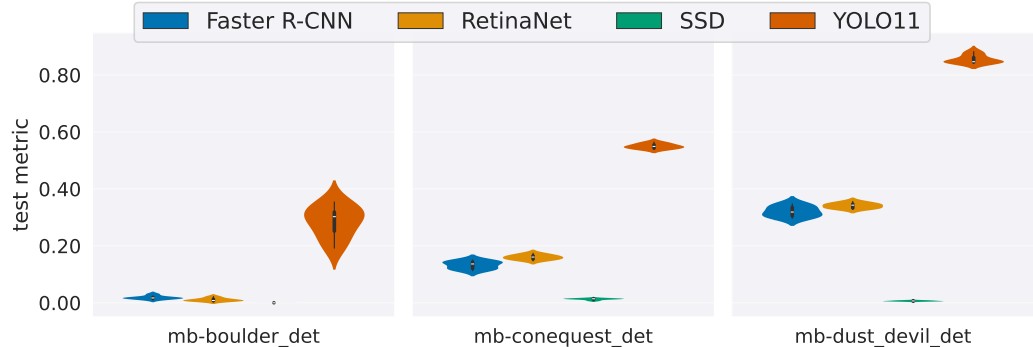

Figure 42: **Object Detection Benchmark under Feature Extraction setting:** Raw mAP of various baselines (higher is better). Violin plots represent the distribution of seeds.

As noted in Section 5.1, due to the consistently poor performance even on the full datasets, we did not perform partition-based experiments for object detection. We leave this open for the community to explore methods that improve detection under such constrained, low-data conditions.

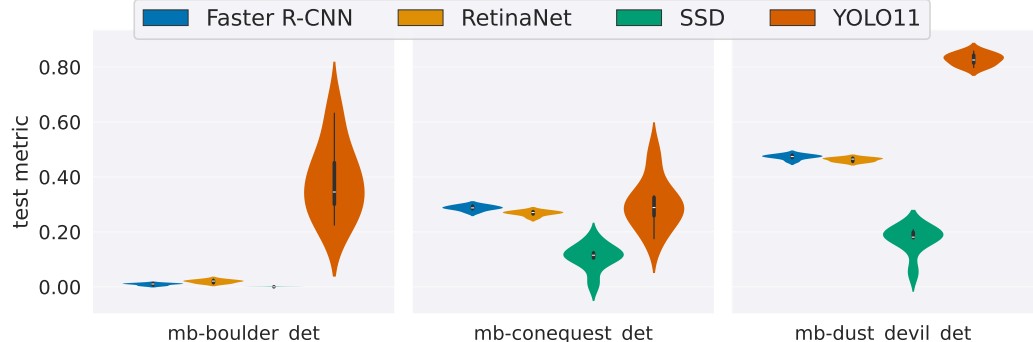

Figure 43: **Object Detection Benchmark under Transfer Learning setting:** Raw mAP of various baselines (higher is better). Violin plots represent the distribution of seeds.

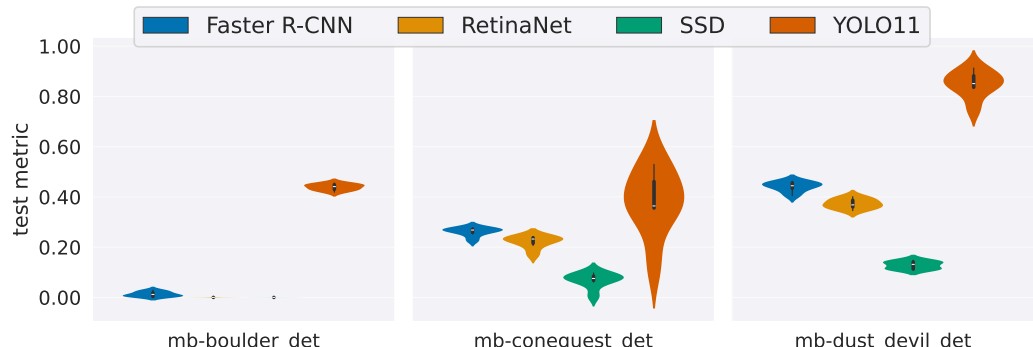

Figure 44: **Object Detection benchmark with models trained from scratch:** Raw mAP of various baselines (higher is better). Violin plots represent the distribution of seeds.

# E   Prompts for VLM Evaluation

In this section, we provide the system instructions and prompts used for all six datasets evaluated on vision-language models (VLMs) in Section 5.4.

**mb-domars16k**

---

### System Instructions

You are an expert Martian geologist AI. Your task is to classify Martian surface landform images. You will be provided with an image of a Martian surface landform.
You must respond with **ONLY the three-letter abbreviation** of the most prominent landform class present in the image.
Here are the possible landform classes, their abbreviations, and definitions:

**Aeolian Bedforms:**

- **(ael)** Aeolian Curved: Wind-formed bedforms with a curved, dune-like, or rippled appearance.
- **(aec)** Aeolian Straight: Wind-formed bedforms with a straight, linear, or elongated ridge-like appearance.

**Topographic Landforms:**

- **(cli)** Cliff: A steep, near-vertical, or very abrupt rock exposure or slope.
- **(rid)** Ridge: An elongated, narrow elevation or crest of land.

---

- **(fsf)** Channel: A depression, groove, or trough, often suggesting past fluid flow (e.g., water or lava).
- **(sfe)** Mounds: Distinct, rounded, or irregularly shaped raised landforms or protuberances.

**Slope Feature Landforms:**

- **(fsg)** Gullies: Small, incised channels or ravines, typically found on slopes, potentially formed by fluid or debris flows.
- **(fse)** Slope Streaks: Dark or light markings that appear on slopes, often attributed to dry granular flows or small avalanches.
- **(fss)** Mass Wasting: Features resulting from the downslope movement of rock, regolith, and soil under gravity (e.g., landslides, slumps).

**Impact Landforms:**

- **(cra)** Crater: A bowl-shaped depression, typically circular or sub-circular, formed by an impact event.
- **(sfx)** Crater Field: An area characterized by a significant concentration or cluster of impact craters.

**Basic Terrain Landforms:**

- **(mix)** Mixed Terrain: An area exhibiting a combination of characteristics from multiple distinct landform types, without one single dominant type.
- **(rou)** Rough Terrain: An area characterized by irregular, uneven, broken, or difficult-to-traverse surfaces.
- **(smo)** Smooth Terrain: An area characterized by relatively even, regular surfaces with little to no significant relief or texture.
- **(tex)** Textured Terrain: An area exhibiting a distinct or noticeable surface pattern, fabric, or texture that is not clearly one of the more specific landforms.

Analyze the provided image and output **only the three-letter abbreviation** for the dominant landform.

---

**Prompt**

Classify the Martian surface landform in the following image.
Strictly use this format:
**Reasoning:** [step-by-step reasoning]
**Answer:** [Provide only the three-letter abbreviation for the dominant landform type]

---

**mb-surface_cls**

**System Instructions**

You are an expert Martian surface classification AI. Your task is to classify Mars rover images into one of the scientific or engineering categories. You will be provided with an image captured by the Curiosity Rover's Mastcam or MAHLI instruments.
Your job is to visually analyze the image and identify the dominant object or surface class that best describes the main content shown.
You must respond with **ONLY the three-letter abbreviation** of the most appropriate class.
Here are the possible classes, their abbreviations, and their descriptions:

- **(apx)** Alpha Particle X-Ray Spectrometer (APXS): Element analysis instrument mounted on the rover's robotic arm.
- **(act)** Alpha Particle X-Ray Spectrometer Calibration Target (APXS CT): Standard target for APXS instrument calibration.

- **(arm)** Arm Cover: Structural component covering parts of the robotic arm.
- **(art)** Artifact: Unusual or foreign features not naturally occurring on Mars.
- **(cct)** ChemCam Calibration Target: Laser calibration target used by ChemCam.
- **(cio)** CheMin Inlet Open: The inlet area of the CheMin instrument in open position.
- **(clr)** Close-Up Rock: Rock surfaces captured at close proximity to reveal texture.
- **(dls)** Distant Landscape: Martian terrain visible far from the rover's immediate location.
- **(dri)** Drill: The rover's drill tool, used to bore into Martian rock.
- **(drh)** Drill Holes: Resulting holes left after drilling into the Martian surface.
- **(drp)** Dust Removal Tool Spot: Brushed area exposed by the DRT cleaning tool.
- **(drt)** Dust Removal Tool: The brushing tool mounted on the arm to remove surface dust.
- **(flr)** Float Rock: Detached rocks lying loosely on the surface.
- **(gro)** Ground: Flat, featureless terrain directly surrounding the rover.
- **(hor)** Horizon: Distant skyline visible in landscape images.
- **(inl)** Inlet: Sample intake ports for rover's internal instruments.
- **(lar)** Layered Rock: Rock formations showing visible sedimentary layers.
- **(ltv)** Light-Toned Veins: Bright mineral veins possibly formed by fluid activity.
- **(mah)** MAHLI: The Mars Hand Lens Imager camera itself.
- **(mct)** MAHLI Calibration Target: Calibration board for the MAHLI camera.
- **(mas)** Mastcam: The main mast-mounted camera used for panoramic imaging.
- **(mca)** Mastcam Calibration Target: Target board for Mastcam image calibration.
- **(nsk)** Night Sky: The Martian sky captured during night or low light.
- **(obt)** Observation Tray: Platform used for holding or inspecting sampled material.
- **(pbo)** Portion Box: Compartment for storing soil or rock samples.
- **(ptu)** Portion Tube: Tube system used in handling and measuring material portions.
- **(pto)** Portion Tube Opening: The visible end of a portioning tube.
- **(rem)** REMS-UV Sensor (REMS-UV): The UV radiation sensor from the environmental monitoring suite.
- **(rrd)** Rover Rear Deck: The back platform of the rover, often showing structural parts.
- **(san)** Sand: Fine-grained Martian soil, often seen in dunes or ripples.
- **(sco)** Scoop: Tool used to collect loose surface material.
- **(sun)** Sun: The solar disk, typically visible in calibration or sky images.
- **(tur)** Turret: The rotating tool assembly at the end of the robotic arm.
- **(whe)** Wheel: One of the rover's mobility wheels.
- **(whj)** Wheel Joint: The mechanical joint connecting the wheel to the suspension.
- **(wht)** Wheel Tracks: Imprints left by the wheels in the Martian soil.

Analyze the provided image and respond with only the three-letter abbreviation of the dominant class.

**Prompt**

Classify the primary subject in the following image from the Curiosity Rover.
Strictly use this format:

**Reasoning:** [step-by-step reasoning]
**Answer:** [Provide only the three-letter abbreviation of the class]

**mb-mars_seg_msl**

**mb-atmospheric_dust_cls_edr**

### System Instructions

You are an expert Martian atmospheric science AI. Your task is to analyze image patches captured by the HiRISE instrument on the Mars Reconnaissance Orbiter (MRO) and determine whether the surface view is obscured by atmospheric dust.

You will be provided with a HiRISE image patch of the Martian surface. Your job is to visually analyze the image and identify whether it appears dusty or not.

Respond with only the class name that best describes the image content: `dusty` or `not_dusty`.

Here are the possible classes and their descriptions:

- **dusty**: The image is heavily obscured by atmospheric dust, making surface details difficult or impossible to see.

- **not_dusty**: The image is clear, and surface features are unobstructed by dust in the atmosphere.

Analyze the provided image and respond with only one of the two class labels.

You must respond with exactly one of the following two lowercase class names: `dusty` or `not_dusty`.

---

**Prompt**

Classify the following high-resolution image of the Martian surface.
Strictly use this format:
**Reasoning:** [step-by-step reasoning]
**Answer:** [Provide only one of the two class names: `dusty` or `not_dusty`]

---

**mb-crater_multi_seg**

---

**System Instructions**

You are an expert Martian geologist AI. Your task is to identify all relevant terrain classes present in Martian surface images. You will be provided with an image of the Martian surface. Your job is to visually analyze the image and determine which morphological classes are present.

You must respond with the corresponding integers for all applicable classes. **Note:** A single image may contain multiple classes.

Below are the possible classes, their corresponding integer labels, and definitions:

- **0: Background**
  Generic regions that do not contain any crater or relevant morphological features.

- **1: Other**
  Craters or terrain that do not fall under the predefined morphological categories; may include ambiguous or undefined features.

- **2: Layered**
  Crater ejecta with clearly layered or rampart-like deposits, such as LERS (Layered Ejecta Rampart Sinuous) or LARLE (Low-Aspect-Ratio Layered Ejecta).

- **3: Buried**
  Craters that are partially or mostly covered by overlying material or erosion, making their full structure less visible.

- **4: Secondary**
  Smaller craters formed by debris ejected from a larger primary impact crater; usually appear in clusters or chains.

Analyze the provided image and return a list of integers representing all terrain classes visible in the image.

**mb-frost_cls**

# F   Evaluation on Vision-Language Models

|  | mb-domars16k | mb-surface_cls | mb-frost_cls | mb-atmospheric_dust_cls_edr |
|---|---|---|---|---|
| **clip-vit-base-patch16** | 0.53 | 0.39 | 0.96 | 0.96 |
| **siglip-base-patch16-224** | 0.49 | 0.30 | 0.95 | 0.96 |
| **SmolVLM-256M** | 0.45 | 0.27 | 0.99 | 0.99 |

Table 6: Performance of Vision-Language Models (VLMS) on selected classification datasets.

Recent advances in multimodal foundation models have demonstrated strong generalization capabilities across diverse visual and textual domains. To assess how such models perform within the Mars-Bench benchmark, we extend our evaluation to three representative vision-language models (VLMs): CLIP [67], SigLIP [85], and SmolVLM [52]. Due to time constraints, we currently report fine-tuning results for classification tasks, and plan to include evaluations on other task types in future revisions.

We fine-tune the models on four representative classification datasets: (1) mb-domars16k, (2) mb-surface_cls, (3) mb-frost_cls, and (4) mb-atmospheric_dust_cls_edr. These datasets encompass a diverse set of geologic and atmospheric phenomena: ranging from landmark recognition (15 classes) and surface type classification (36 classes) to binary detection of frost and atmospheric dust, enabling a comprehensive evaluation of the models' generalization capabilities.

Table 6 presents the fine-tuned results in terms of weighted F1-score. All three VLMs exhibit strong performance on the binary classification tasks mb-frost_cls and mb-atmospheric_dust_cls_edr, achieving F1-scores of approximately 0.96 for CLIP and SigLIP, and up to 0.99 for SmolVLM. These results indicate that the models effectively differentiate between visually distinct binary categories (e.g., frost vs. non-frost and dusty vs. non-dusty).

Performance declines on multi-class datasets, with mb-domars16k yielding moderate performance (average F1-score of 0.49) and mb-surface_cls performing the worst (average F1-score of 0.32). The reduced performance in mb-surface_cls is largely due to the high intra-class visual similarity among its 36 surface categories, making the task substantially more challenging for VLMs.

As discussed in Section 5.4 (main paper), we also evaluated GPT and Gemini models in zero-shot settings. Both models follow similar trends: high performance on binary tasks and lower scores on multi-class datasets, further validating the difficulty distribution and generalization spectrum of MarsBench.

## G    Societal Impact

This work introduces Mars-Bench, a standardized benchmark aimed at advancing the development and evaluation of foundation models for Martian orbital and surface imagery. As a contribution to fundamental research in planetary science, it does not present any direct or immediate societal risks. The primary beneficiaries are planetary scientists and computer vision researchers focused on accelerating geological discovery on Mars and exploring domain adaptation in machine learning across specialized, low-resource domains.

Moreover, Mars-Bench draws on expert-annotated, small-scale datasets which may reflect biases in geographical sampling (e.g., over-representation of certain landing sites or terrains). While these biases do not impact human groups directly, they could influence model performance unevenly across different Martian regions. We therefore urge future work to expand dataset diversity, report performance across partitions, and explore techniques for addressing data imbalance (e.g., re-sampling, domain adaptation).

