# OpenReview forum: "Mars-Bench: A Benchmark for Evaluating Foundation Models for Mars Science Tasks"
_NeurIPS.cc/2025/Datasets_and_Benchmarks_Track — NeurIPS 2025 Datasets and Benchmarks Track poster_

### Official Review · Reviewer_qJyk · 2025-06-28

**Rating:** 4
**Confidence:** 4

**Summary:**

This paper introduces a comprehensive benchmark for research on Mars, covering tasks such as classification, segmentation, and object detection. It provides user-friendly tools and standardized evaluation protocols to facilitate experimentation. The benchmark also includes experiments on few-shot learning and domain-adaptive learning scenarios, accompanied by thoughtful discussions.

**Dataset Code Accessibility:**

Yes

**Ethical Considerations:**

No, there are no or only very minor ethics concerns

**Final Justification:**

Thank you for the response. It addresses most of my concerns. However, I believe the important results—particularly those on object detection—should be included in the main paper rather than in the supplementary material. I would keep my initial score.

**Limitations Weaknesses:**

(1) Limited Data Novelty: All datasets included in the benchmark are sourced from existing publications. While consolidation can be valuable, the lack of newly collected or annotated data may limit the paper’s originality and its contribution to the research community.

(2) Missing Experimental Coverage: Although the paper references benchmarks for object detection, there appear to be no corresponding experimental results or analysis provided. This disconnect weakens the completeness of the benchmark evaluation.

(3) Experimental Section Lacks Depth: The experimental setup could be significantly expanded. For example, there are no results reported for training from scratch on the Mars-benchmark, and the range of evaluated methods is fairly limited. Incorporating more recent and powerful approaches—such as state-of-the-art multimodal foundation models for segmentation and detection—would strengthen the evaluation and provide a more comprehensive assessment of model performance.

**Strengths Contributions:**

(1) The motivation is engaging, particularly in highlighting the contrast between Earth Observation (EO)-based images and Mars-based imagery. This distinction effectively sets the stage for exploring the unique challenges in the domain.

(2) The provision of tools and standardized evaluation protocols is valuable, as it facilitates fair and reproducible comparisons across different baseline methods, contributing to future benchmarking efforts.

(3) The experimental section is clearly presented, addressing key questions that are likely to arise from readers and offering insights that enhance the overall understanding of the model’s behavior and performance.

---

> ### Author Rebuttal · Authors · 2025-07-31
>
> Thank you for your time and great feedback. We appreciate that the reviewer acknowledges extensive evaluation and presentation.
>
> > ***[1]*** Limited Data Novelty: All datasets included in the benchmark are sourced from existing publications. While consolidation can be valuable, the lack of newly collected or annotated data may limit the paper’s originality and its contribution to the research community.
>
> Thank you for this comment. We would like to clarify that although there are many open-source datasets available for Mars-related tasks, they are not in a ready-to-use format. And specialized domains (planetary science in our case), data processing can take a lot of time and may require expert supervision. With this, many of these open-source datasets do not provide train/validation/test splits, which makes it challenging for reproducibility and to compare across different methods.
>
> Also, many such benchmarks (similar to Mars-Bench) have played a critical role in driving progress, especially in specialized domains such as medical, earth observation, and astronomy. And currently, no unified benchmark exists for Mars science applications. Mars-Bench fills this gap by providing a ready-to-use, open-source benchmark with standardized train/validation/test splits and consistent formats across diverse datasets. By consolidating and organizing datasets, Mars-Bench reduces the time required for data curation and preprocessing, benefiting the broader community and enabling faster progress.
>
>
> > ***[2]*** Missing Experimental Coverage: Although the paper references benchmarks for object detection, there appear to be no corresponding experimental results or analysis provided. This disconnect weakens the completeness of the benchmark evaluation.
>
> We would like to mention that we did conduct experiments for object detection and included the results in the paper. Due to space constraints in the main manuscript, we added object detection results in the supplementary material. Specifically, please refer to Section 5, line no. 204 and Appendix C.3 for object detection results.
>
>
> > ***[3]*** Experimental Section Lacks Depth: The experimental setup could be significantly expanded. For example, there are no results reported for training from scratch on the Mars-benchmark, and the range of evaluated methods is fairly limited. Incorporating more recent and powerful approaches—such as state-of-the-art multimodal foundation models for segmentation and detection—would strengthen the evaluation and provide a more comprehensive assessment of model performance.
>
> We would like to clarify that we did conduct experiments where we trained a model from scratch (weights are randomly initialized) on all datasets, and we reported results in the paper. Please refer to Section 4, line no. 182 and Appendix C.
>
> For the second part of the comment, we thank the reviewer for this feedback. We agree that fine-tuning vision-language models (VLMs) on Mars-Bench datasets can further strengthen our paper.
>
> We conducted experiments on 3 well-known VLMs: CLIP, SigLIP, and SmolVLM. Due to the limited rebuttal time period, currently, we only show results from classification tasks; we’ll show evaluation on other tasks in the revised version.
>
> We show fine-tuning results on 4 classification datasets: (1) mb-domars16k, (2) mb-surface_cls, (3) mb-frost_cls, and (4) mb-atmospheric_dust_cls_edr. The tasks are selected in a way that covers a range of geologic features (i.e., 15 landmark classes, 36 surface classes, and frost and atmospheric dust classes) to evaluate how well the models generalize across different scientific concepts.
>
> We show the results (F1-score) in the table below:
>
> |                  | mb-domars16k | mb-surface_cls | mb-frost_cls | mb-atmospheric_dust_cls_edr |
> |:------------------------:|:--------------:|:----------------:|:--------------:|:-----------------------------:|
> | CLIP (ViT-B/16)        | 0.53         | 0.39           | 0.96         | 0.96                         |
> | SigLIP (B/16-224)      | 0.49         | 0.30           | 0.95         | 0.96                         |
> | SmolVLM-256M           | 0.45         | 0.27           | 0.99      | 0.99                      |
>
> Results show that all three VLMs show strong performance on the binary classification tasks mb-frost_cls and mb-atmospheric_dust_cls_edr; CLIP and SigLIP show ~0.96 F1-score, while SmolVLM shows 0.99 F1-score. The reason behind this is that these tasks are simpler, distinguishing between two visually distinct classes (i.e., frost vs. non-frost and dusty vs. non-dusty).
>
> In contrast, mb-domars16k yields moderate performance with an average F1-score of 0.49, while mb-surface_cls performs the worst (average F1-score 0.32). The reduced performance in mb-surface_cls is primarily due to the high intra-class visual similarity among its 36 surface categories, which introduces a greater challenge for VLMs.
>
> These trends align with the results that we have shown in the paper. Specifically, we finetune and evaluate these datasets on the ImageNet pre-trained models (Section 5.1), models pre-trained on Earth satellite data (Section 5.3); and we also evaluate GPT and Gemini in _zero-shot settings_ (Section 5.4). Results from all these models show strong performance on binary tasks and comparatively lower on multi-class datasets, which further validates the difficulty distribution across the benchmark. To iterate, this observation also aligns with results on VLMs finetuning.
>
> We will include these updated findings in the revised version of our paper.

---

> > ### Author Response · Authors · 2025-08-05
> >
> > We thank the reviewer for their time and effort in providing feedback for our paper.
> >
> > We have addressed all your questions and suggestions in the rebuttal. As we approach the end of the author-reviewer discussion period, please let us know if you have any additional comments regarding our responses.

---

> > ### Comment · Reviewer_qJyk · 2025-08-05
> >
> > Thank you for the response. It addresses most of my concerns. However, I believe the important results—particularly those on object detection—should be included in the main paper rather than in the supplementary material.

---

> > > ### Author Response · Authors · 2025-08-05
> > >
> > > Thank you for your response! We agree with your suggestion to include the object detection results in the main paper and will incorporate them in the revised version. We hope this addresses all of your concerns.

---

> > ### Author Response · Authors · 2025-08-08
> >
> > Hello **qJyk**,
> >
> > We have carefully addressed all your comments and suggestions. As we are approaching the end of the author-reviewer discussion period, we would greatly appreciate it if you could kindly review the rebuttal.

---

### Official Review · Reviewer_EQgW · 2025-07-01

**Rating:** 5
**Confidence:** 4

**Summary:**

The submission introduces a new large-scale image benchmark (Mars-Bench) to assess the progress of computer vision research in the field of Mars Science. By repurposing existing datasets (captured by different sensors such as Mars orbiters, rovers, and so on) and working with domain experts, the authors are able to create a benchmark consisting of 20 tasks with the same standardized format, allowing systematic and standardized evaluation of future work. On the benchmark, the authors have tested multiple interesting settings, such as (1) training from scratch, (2) probing feature extractors, and (3) fine-tuning pre-trained models. Extensive and rigorous experiments have yielded interesting results, such as the finding that ImageNet models are more suitable for Mars Science compared to models pretrained on Earth Science satellite imagery. Additionally, it has been shown that general-purpose multimodal large language models are not equipped to handle Mars Science. Overall, this is an interesting benchmark that could be of interest to the community and facilitates more interesting research directions.

**Additional Feedback:**

The reviewer would appreciate it if the authors could address W2.

**Dataset Code Accessibility:**

Yes

**Dataset Code Comments:**

- The dataset could be visualized on Huggingface
- The authors have provided a GitHub link for training all the models and for future work to build upon.

**Ethical Considerations:**

No, there are no or only very minor ethics concerns

**Final Justification:**

The reviewer's major concern has been resolved (no clear mention of the expert's background) during the rebuttal period, thus deciding to accept the paper.

**Limitations Weaknesses:**

[W1] Lack of novelty in the data curation process: the benchmark is a coalescence of multiple existing datasets. While the authors work with domain experts to validate the annotation, the process is hardly novel. However, the reviewer is willing to let this go, given the interesting domain and results.
[W2] No mention of the background of experts: the reviewer was not able to find any information about the experts who verified the annotations. While the authors mentioned that domain experts collect each of the individual datasets, it is still important to note the background of the validators (or the training they have received before validation).

**Strengths Contributions:**

[S1] Exciting domain: With the large amount of unlabeled data and different sensors available, the benchmark can open up interesting new research directions, such as sensor fusion, unsupervised pre-training, domain adaptation, and training multimodal large language models for Mars Science
[S2] Extensive experiments: Hypotheses are properly tested, and results are reported with statistical rigor
[S3] Presentation: The paper is well-written and easy to digest.
[S4] Interesting results: ImageNet being a more suitable pre-training domain than satellite imagery from Earth sciences is noteworthy and warrants further investigation.

---

> ### Author Rebuttal · Authors · 2025-07-31
>
> Thank you for your time and thoughtful feedback. We are glad that the reviewer finds the domain exciting and acknowledges our extensive experiments, results, and presentation.
>
> > ***[W1]*** _Lack of novelty in the data curation process: the benchmark is a coalescence of multiple existing datasets. While the authors work with domain experts to validate the annotation, the process is hardly novel. However, the reviewer is willing to let this go, given the interesting domain and results._
>
> Thank you for this comment. We would like to clarify that although there are many open-source datasets available for Mars-related tasks, they are not in a ready-to-use format. And specialized domains (planetary science in our case), data processing can take a lot of time and may require expert supervision. With this, many of these open-source datasets do not provide train/validation/test splits, which makes it challenging for reproducibility and to compare across different methods.
>
> Also, many such benchmarks (similar to Mars-Bench) have played a critical role in driving progress, especially in specialized domains such as medical [1, 2, 3], earth observation [4], and law [5]. And currently, no unified benchmark exists for Mars science applications. Mars-Bench fills this gap by providing a ready-to-use, open-source benchmark with standardized train/validation/test splits and consistent formats across diverse datasets. By consolidating and organizing datasets, Mars-Bench reduces the time required for data curation and preprocessing, benefiting the broader community and enabling faster progress.
>
> > ***[W2]*** _No mention of the background of experts: the reviewer was not able to find any information about the experts who verified the annotations. While the authors mentioned that domain experts collect each of the individual datasets, it is still important to note the background of the validators (or the training they have received before validation)._
>
> We appreciate the reviewer’s feedback and agree that including background information of experts can strengthen the validity of Mars-Bench.
>
> Co-author Jacob Adler is a geologist and planetary science professor who has worked on Mars science for the past 15 years. Co-author Umaa Rebbapragada is the technical group supervisor of the Machine Learning and Instrument Autonomy team at NASA’s Jet Propulsion Laboratory (JPL), which has developed machine learning solutions for numerous planetary science studies and missions., Co-author Steven Lu is a data scientist on the same team at JPL; he has supported numerous planetary science missions and is the Technologist for NASA's Planetary Data System Imaging and Cartography Sciences Node (PDS Imaging Node), which distributes image data for NASA missions. These co-authors contributed significant domain expertise for selecting datasets and tasks, as well as validating and verifying datasets.
>
> We’ll include this information in the revised version.
>
> ---
>
> [1] Parmar M., Mishra S., Purohit M., Luo M., Mohammad M., and Baral C. "In-BoXBART: Get Instructions into Biomedical Multi-Task Learning", *NAACL* 2022.
>
> [2] Fries J., Weber L., Seelam N., Altay G., Datta D., Garda S., Kang S., Su R., Kusa W., and Cahyawijaya S. "BigBio: A framework for data-centric biomedical NLP", *NeurIPS* 2022.
>
> [3] Johnson A.E.W., Bulgarelli L., Shen L., Gayles A., Shammout A., Horng S., Pollard T.J., Hao S., Moody B., and Gow B. "MIMIC-IV: A freely accessible electronic health record dataset", *Scientific Data* 2023.
>
> [4] Lacoste, A., Lehmann, N., Rodriguez, P., Sherwin, E., Kerner, H., Lütjens, B., Irvin, J., Dao, D., Alemohammad, H., Drouin, A., & others. "Geo-Bench: Toward foundation models for Earth monitoring", *NeurIPS* 2023.
>
> [5] Guha, N., Nyarko, J., Ho, D., Ré, C., Chilton, A., Chohlas-Wood, A., Peters, A., Waldon, B., Rockmore, D., Zambrano, D., & others. "LegalBench: A collaboratively built benchmark for measuring legal reasoning in large language models", NeurIPS 2023.

---

> > ### Author Response · Authors · 2025-08-05
> >
> > We thank the reviewer for their time and effort in providing feedback for our paper.
> >
> > We have addressed all your questions and suggestions in the rebuttal. As we approach the end of the author-reviewer discussion period, please let us know if you have any additional comments regarding our responses.

---

> > > ### Comment · Reviewer_EQgW · 2025-08-06
> > > **Thank you for the rebuttal!**
> > >
> > > The reviewer thanks the authors for the rebuttal. The reviewer's concerns have been sufficiently resolved, and will keep the rating as-is.

---

### Official Review · Reviewer_2yNW · 2025-07-03

**Rating:** 5
**Confidence:** 3

**Summary:**

This dataset presents data samples specifically for Mars science tasks, which has very unique patterns and challenges. The tasks include most general vision and perception tasks such as classification, segmentation and object detection, with additional challenges due to special geology patterns. The collection of the dataset involves multiple scientists of diverse fields for data cleaning, annotations, etc. Multiple open-source tools are provided, including visualization utils, reproduciable code and baselines. Most state-of-the-art models are evaluated (e.g., ViT-L/16, SwinV2-B, U-Net, YOLO, RetinaNet, as well as VLMs) under different training strategy to estimate each model's capability. Domain transfer from earth data to Mars data is also carried out, to reveal the domain generalization capability.

**Dataset Code Accessibility:**

Yes

**Dataset Code Comments:**

The dataset and code are accessible to me with good documentation provided. I think the code contains enough details for reproducibility.

**Ethical Comments:**

I think this dataset has only minor (if not none) ethics issues.

**Ethical Considerations:**

No, there are no or only very minor ethics concerns

**Final Justification:**

After reading through authors' feedback and other reviews, I decided to keep my rating.

**Limitations Weaknesses:**

- The dataset still miss some information, such as the geological location, making it difficult to test the generalization and domain transferability for different major geological areas.

- Class imbalance: the benchmark contains much less instances of dust devils or/and cones, which can lead to class imbalance and unreliable or extreme evaluation. Data augmentation, reweighting and other similar strategies are not employed to reduce the effect of class imbalance.

- Most experiments rely on models pretrained on earth data. What if the models are pretrained on Mars data?

**Strengths Contributions:**

- This benchmark has evaluated most state-of-the-art models in a fairly thorough manner (e.g., with different training strategy and on different tasks).

- The data samples are of great uniqueness and provide complementary / different information compared to similar data collected on earth.

---

> ### Author Rebuttal · Authors · 2025-07-31
>
> Thank you for your time and great feedback. We appreciate that the reviewer acknowledges the comprehensive evaluation of the benchmark and the uniqueness of the data samples.
>
> > ***[1]*** _The dataset still miss some information, such as the geological location, making it difficult to test the generalization and domain transferability for different major geological areas._
>
> We agree with the reviewer’s comment that geological information is very crucial in any remote sensing dataset, as it helps in evaluating generalization and domain transferability across different geologic regions. However, many of the original datasets we include in Mars-Bench did not provide this information, and it cannot be retrieved from the available file identifiers alone. We clarified this limitation in Section 7, line 342.
>
> In addition, we have included georeferenced samples wherever possible. For example, mb-conequest contains geolocation metadata as it was provided in the original dataset. In the case of the mb-crater_seg datasets, the data samples were generated by us from the original satellite tiles. With this, if there are geo-referenced datasets published in the future, we look forward to incorporating them into Mars-bench.
>
>
> > ***[2]*** _Class imbalance: the benchmark contains much less instances of dust devils or/and cones, which can lead to class imbalance and unreliable or extreme evaluation. Data augmentation, reweighting and other similar strategies are not employed to reduce the effect of class imbalance._
>
> Thank you for the suggestion. We agree that data manipulation techniques can help to overcome the data imbalance issue. Based on the suggestion, we conducted additional experiments with the data manipulation techniques that help reduce the effect of class imbalance.
>
> We applied three commonly used techniques: (1) undersampling, (2) oversampling, and (3) loss weighting, across four highly imbalanced datasets from Mars-Bench: (1) mb-landmark_cls, (2) mb-surface_cls, (3) mb-change_cls_ctx, and (4) mb-change_cls_hirise. Experiments were conducted under feature extraction settings using five classification models originally included in our paper: SqueezeNet, ResNet, Inception, Vision Transformer (ViT), and Swin Transformer (SwinT). To evaluate the impact, we compared each technique against the baseline (standard training without any data manipulation).
>
> Due to space constraints, we present average results (F1-score) across models for each technique; however, we will include full results in the revised version.
>
> |             | mb-landmark_cls | mb-surface_cls | mb-change_cls_ctx | mb-change_cls_hirise |
> |:---------------------:|:-----------------:|:----------------:|:-------------------:|:----------------------:|
> | Standard            | 0.84            | 0.77           | 0.78              | 0.86                 |
> | Oversampling        | 0.60            | 0.54           | 0.54              | 0.76                 |
> | Undersampling       | 0.41            | 0.34           | 0.58              | 0.79                 |
> | Loss weighting      | 0.68            | 0.58           | 0.68              | 0.81                 |
>
> - **Overall performance:** From the table, it can be observed that while data manipulation techniques help balance classes, they often result in a decrease in overall accuracy compared to the standard setup. Among the techniques, loss weighting generally maintains better performance, but even it does not consistently outperform the standard baseline across all scenarios.
>
> However, the aggregated results do not provide a complete picture of how each technique impacts the performance of the minority and majority classes individually. To interpret these results better, we analyzed the class-wise performance by comparing each data manipulation technique with the baseline (standard setup):
>
> - **Loss weighting:** Same as aggregated results, loss weighting is the most consistent technique in improving minority class performance across all datasets and models, while also maintaining relatively stable performance for the majority class.
>     - Example: In the mb-change_cls_hirise dataset using ResNet, the minority class ($\texttt{Change}$) accuracy improved from 0.17 to 0.48, while the majority class ($\texttt{No Change}$) performance dropped only slightly from 0.96 to 0.94. Similarly, in the mb-surface_cls dataset, the minority class ($\texttt{Arm Cover}$) improved from 0.00 to 0.50 with ResNet.
>
> - **Undersampling:** Although balancing data, a decreased number of training samples shows a negative effect on performance. Performance for the minority class does not show significant improvement and significantly degrades the performance of the majority class, particularly for transformer-based models, which typically require more data to converge.
>     - Example: In mb-landmark_cls, the minority class ($\texttt{Impact Ejecta}$) improved marginally from 0.00 to 0.03 with Inception, while the majority class ($\texttt{Others}$) accuracy dropped from 0.93 to 0.35. A similar trend was observed in mb-surface_cls, where the minority class ($\texttt{Arm Cover}$) improved only from 0.00 to 0.01, while the majority class ($\texttt{Ground}$) performance dropped from 0.72 to 0.03.
>
> - **Oversampling:** Oversampling shows mixed results. It shows significant effectiveness in transformer-based models but is less consistent in convolutional models.
>     - Example: In the mb-change_cls_hirise dataset, ViT showed improvement in the minority class from 0.39 to 0.52, while the majority class remained almost unaffected (0.96 to 0.95). SwinT shows similarly stable behavior across datasets. In contrast, ResNet benefited from oversampling in some datasets (e.g., from 0.00 to 0.62 in mb-surface_cls and 0.17 to 0.35 in mb-change_hirise). However, in SqueezeNet, although it retains the performance of the minority class, it shows performance degradation in the majority class, dropping from 0.84 to 0.51 in mb-surface_cls and from 0.32 to 0.18 in mb-change_cls_hirise.
>
> In summary, all these results indicate that there is no single data manipulation technique that is universally effective across all model and dataset combinations for handling class imbalance. This suggests that the choice of technique depends heavily on the specific characteristics of the dataset and the model being used. We believe this presents a significant opportunity for the community to develop more specialized solutions for imbalanced data in niche domains like planetary science, where class distributions can be highly diverse.
>
> We will include these updated findings in the revised version of our paper.
>
> > ***[3]*** _Most experiments rely on models pretrained on earth data. What if the models are pretrained on Mars data?_
>
> We agree with the reviewer that fine-tuning on a model which is pre-trained on Mars data can show interesting results and give very different results compared to ImageNet or Earth-satellite pre-training. However, there are _no_ pre-trained Mars model weights available. Additionally, pre-training a model on Mars data can be challenging, as exporting satellite data, followed by processing and filtering, is a time-consuming and resource-intensive process, which is beyond the scope of this work.
>
> And as we mentioned in the Introduction, we believe that Mars-Bench will enable building Mars-specific foundation models that can be evaluated on tasks from Mars-Bench.

---

> > ### Author Response · Authors · 2025-08-05
> >
> > We thank the reviewer for their time and effort in providing feedback for our paper.
> >
> > We have addressed all your questions and suggestions in the rebuttal. As we approach the end of the author-reviewer discussion period, please let us know if you have any additional comments regarding our responses.

---

> > > ### Comment · Area_Chair_1g1n · 2025-08-06
> > >
> > > Hi **2yNW**,
> > >
> > > Can I ask you to please engage with the discussion before the deadline this week.
> > >
> > > Thanks,
> > > Your AC

---

> > > > ### Author Response · Authors · 2025-08-08
> > > >
> > > > Hello **2yNW**,
> > > >
> > > > We have carefully addressed all your comments and suggestions. As we are approaching the end of the author-reviewer discussion period, we would greatly appreciate it if you could kindly review the rebuttal.

---

> > ### Comment · Reviewer_2yNW · 2025-08-08
> > **feedback**
> >
> > Thanks for the authors' thoughtful and resourceful feedback. Most of my concerns are addressed.
> >
> > Regarding the second issue (data augmentation and class imbalance):
> > - For highly imbalanced datasets, it is recommended (at least in the setting of autonomous driving) to use the average of per-class accuracy rather than the per-sample accuracy, since the latter can be biased towards major classes.
> > - From the numbers presented by the authors, I think the loss weighting method already worked --- just need to show it in the correct "sense".
> > - The other two methods probably needs more design as augmenting data takes a lot of engineering works. I suggest the authors to take more insights from other vision tasks that highly relies on data augmentation. Since each task needs a bit different augmentation pipeline, I believe the augmentation techniques here do have to be tailored.
> >
> > I'm strongly leaning towards acceptance for this paper.

---

> > > ### Author Response · Authors · 2025-08-09
> > >
> > > We really appreciate the reviewer's positive feedback and reviewing our rebuttal carefully.
> > >
> > > We are glad that your perspective on the second issue aligns with our experiments and analyses. To reiterate:
> > >
> > > - We agree that per-class accuracy is important for highly imbalanced datasets. This is why we analyzed results on a class-wise basis. However, due to space constraints, we could not include the full breakdown for all datasets in the main table.
> > > - Our results demonstrate that the loss weighting method improves overall performance compared to the other two methods. However, as noted in our rebuttal, the improvement patterns differ when examining class-wise performance, which we also analyzed in detail.
> > > - We share your view that effective data augmentation pipelines must be tailored to specific datasets and tasks. This tailoring requirement was acknowledged in our rebuttal.
> > >
> > > Thank you very much for showing your support for the acceptance of our work.

---

### Official Review · Reviewer_K7Qp · 2025-07-03

**Rating:** 4
**Confidence:** 3

**Summary:**

This paper introduces **Mars-Bench**, the first comprehensive benchmark for evaluating foundation models on Martian science applications. While foundation models and standardized benchmarks have accelerated progress in domains like medical imaging and Earth observation, Martian ML applications have lacked such infrastructure despite the abundance of high-resolution Mars imagery. Mars-Bench addresses this gap by curating and standardizing datasets from orbiters and rovers into a unified, ML-ready format. This enables systematic evaluation of generalization and robustness across diverse Mars-related tasks, fostering reproducibility and scientific discovery in planetary science.

**Dataset Code Accessibility:**

Yes

**Ethical Considerations:**

No, there are no or only very minor ethics concerns

**Final Justification:**

Since the authors have addressed all of my concerns, I will maintain my score. Please ensure that the revised manuscript clearly reflects the contents of the rebuttal.

**Limitations Weaknesses:**

- The paper would benefit from including more illustrative image examples that highlight the unique characteristics of each dataset, which would help readers more intuitively and efficiently understand the dataset composition.

- Given the increasing relevance of vision-language models, presenting fine-tuning results on such models would better demonstrate the applicability and versatility of the proposed benchmark in modern multi-modal learning settings.

**Strengths Contributions:**

- Mars-Bench includes 20 curated datasets spanning classification, segmentation, and object detection tasks. The benchmark reflects real scientific use cases in planetary science, covering geologic features, making it both diverse and scientifically grounded.

- The authors benchmark a range of models—including ImageNet-pretrained models, Earth Observation (EO) models, and proprietary vision-language models like Gemini and GPT—under different training hyperparameter configurations. They also analyze model performance across few-shot and partitioned training settings, enabling robust assessment of generalization and data efficiency.

- The benchmark is released with full code, baseline models, and documentation to support dataset handling and result visualization. These well-documented resources facilitate reproducibility and provide strong foundations for future applications in Mars science.

---

> ### Author Rebuttal · Authors · 2025-07-31
>
> Thank you for your time and thoughtful feedback. We appreciate that the reviewer acknowledges our extensive experimental evaluation efforts.
>
> > ***[1]*** _The paper would benefit from including more illustrative image examples that highlight the unique characteristics of each dataset, which would help readers more intuitively and efficiently understand the dataset composition._
>
> Thank you for the suggestion. We agree that showing illustrative samples highlighting the unique characteristics of each dataset can significantly help readers interpret key features more intuitively.
>
> As in Appendix A.2, we have provided detailed descriptions for all datasets, including task type, targeted geologic features, class structure, and observation modality. Based on your suggestion, to further enhance this section, we will include visual illustrations that showcase representative samples for each dataset. These images will highlight the nature of the tasks and dataset properties.
>
> We'll update this in the revised version.
>
> > ***[2]*** _Given the increasing relevance of vision-language models, presenting fine-tuning results on such models would better demonstrate the applicability and versatility of the proposed benchmark in modern multi-modal learning settings._
>
> Thank you for this feedback. We agree that fine-tuning vision-language models (VLMs) on Mars-Bench datasets can further strengthen our paper.
>
> We conducted experiments on 3 well-known VLMs: CLIP, SigLIP, and SmolVLM. Due to the limited rebuttal time period, currently, we only show results from classification tasks; we’ll show evaluation on other tasks in the revised version.
>
> We show fine-tuning results on 4 classification datasets: (1) mb-domars16k, (2) mb-surface_cls, (3) mb-frost_cls, and (4) mb-atmospheric_dust_cls_edr. The tasks are selected in a way that covers a range of geologic features (i.e., 15 landmark classes, 36 surface classes, and frost and atmospheric dust classes) to evaluate how well the models generalize across different scientific concepts.
>
> We show the results (F1-score) in the table below:
>
> |                  | mb-domars16k | mb-surface_cls | mb-frost_cls | mb-atmospheric_dust_cls_edr |
> |:------------------------:|:--------------:|:----------------:|:--------------:|:-----------------------------:|
> | CLIP (ViT-B/16)        | 0.53         | 0.39           | 0.96         | 0.96                         |
> | SigLIP (B/16-224)      | 0.49         | 0.30           | 0.95         | 0.96                         |
> | SmolVLM-256M           | 0.45         | 0.27           | 0.99      | 0.99                      |
>
> Results show that all three VLMs show strong performance on the binary classification tasks mb-frost_cls and mb-atmospheric_dust_cls_edr; CLIP and SigLIP show ~0.96 F1-score, while SmolVLM shows 0.99 F1-score. The reason behind this is that these tasks are simpler, distinguishing between two visually distinct classes (i.e., frost vs. non-frost and dusty vs. non-dusty).
>
> In contrast, mb-domars16k yields moderate performance with an average F1-score of 0.49, while mb-surface_cls performs the worst (average F1-score 0.32). The reduced performance in mb-surface_cls is primarily due to the high intra-class visual similarity among its 36 surface categories, which introduces a greater challenge for VLMs.
>
> These trends align with the results that we have shown in the paper. Specifically, we finetune and evaluate these datasets on the ImageNet pre-trained models (Section 5.1), models pre-trained on Earth satellite data (Section 5.3); and we also evaluate GPT and Gemini in _zero-shot settings_ (Section 5.4). Results from all these models show strong performance on binary tasks and comparatively lower on multi-class datasets, which further validates the difficulty distribution across the benchmark. To iterate, this observation also aligns with results on VLMs finetuning.
>
> We will include these updated findings in the revised version of our paper.

---

> > ### Author Response · Authors · 2025-08-05
> >
> > We thank the reviewer for their time and effort in providing feedback for our paper.
> >
> > We have addressed all your questions and suggestions in the rebuttal. As we approach the end of the author-reviewer discussion period, please let us know if you have any additional comments regarding our responses.

---

> > > ### Comment · Area_Chair_1g1n · 2025-08-06
> > >
> > > Hi **K7Qp**,
> > >
> > > Can I ask you to please engage with the discussion before the deadline this week.
> > >
> > > Thanks,
> > > Your AC

---

> > ### Author Response · Authors · 2025-08-08
> >
> > Hello **K7Qp**,
> >
> > We have carefully addressed all your comments and suggestions. As we are approaching the end of the author-reviewer discussion period, we would greatly appreciate it if you could kindly review the rebuttal.

---

### Note · Authors · 2025-08-12

We sincerely thank all the reviewers for their valuable feedback on our work. We have carefully addressed every comment and suggestion provided by the reviewers.

We would like to highlight the following points for better evaluation:
- **Reviewer K7Qp:** We have addressed all comments raised by this reviewer; however, the reviewer did not participate in the author-reviewer discussion phase.
- **Reviewer qJyk:** Although we addressed all of the reviewer’s comments, the reviewer did not engage further regarding additional comments, which was just a minor suggestion to move some results from the appendix to the main paper (a suggestion we responded positively to).



Thanks,

Authors

---

### Decision · Program_Chairs · 2025-09-18

**Decision:**

Accept (poster)

**Comment:**

This paper introduces Mars-Bench, a new benchmark for evaluating vision models for mars science tasks. It brings together a range of different vision challenges, spanning classification, detection, and segmentation tasks, from existing datasets.

Some of the identified limitations include:
* Limited novelty in the data curation process, i.e., it consists of multiple existing datasets
* Very strong performance on some of the challenges
* With the exception of the commercial VLMs tested, the other baselines are relatively older
* Lack of georeferencing of the datasets

Overall, despite these limitations from reviewers, there was support for accepting the paper. The authors are strongly encouraged to address the reviewer concerns when updating the paper, e.g., add the new VLM results from the rebuttal, add more illustrative examples from the paper, add the detection results to the main paper, and add the missing discussion of the labelling process.